# The transcription factor CBFB suppresses breast cancer through orchestrating translation and transcription

Navdeep Malik[1], Hualong Yan[1], Nellie Moshkovich[2], Murali Palangat[3], Howard Yang[4], Vanesa Sanchez[5], Zhuo Cai[1], Tyler J. Peat[6], Shunlin Jiang[1], Chengyu Liu[7], Maxwell Lee[4], Beverly A. Mock [6], Stuart H. Yuspa[5], Daniel Larson [3], Lalage M. Wakefield[2] & Jing Huang [1]

Translation and transcription are frequently dysregulated in cancer. These two processes are generally regulated by distinct sets of factors. The *CBFB* gene, which encodes a transcription factor, has recently emerged as a highly mutated driver in a variety of human cancers including breast cancer. Here we report a noncanonical role of CBFB in translation regulation. RNA immunoprecipitation followed by deep sequencing (RIP-seq) reveals that cytoplasmic CBFB binds to hundreds of transcripts and regulates their translation. CBFB binds to mRNAs via hnRNPK and enhances translation through eIF4B, a general translation initiation factor. Interestingly, the *RUNX1* mRNA, which encodes the transcriptional partner of CBFB, is bound and translationally regulated by CBFB. Furthermore, nuclear CBFB/RUNX1 complex transcriptionally represses the oncogenic NOTCH signaling pathway in breast cancer. Thus, our data reveal an unexpected function of CBFB in translation regulation and propose that breast cancer cells evade translation and transcription surveillance simultaneously through downregulating CBFB.

[1] Cancer and Stem Cell Epigenetics Section, Laboratory of Cancer Biology and Genetics, Center for Cancer Research, National Cancer Institute, National Institutes of Health, Bethesda, MD 20892, USA. [2] Cancer Biology of TGF-beta Section, Laboratory of Cancer Biology and Genetics, Center for Cancer Research, National Cancer Institute, National Institutes of Health, Bethesda, MD 20892, USA. [3] Laboratory of Receptor Biology & Gene Expression, Center for Cancer Research, National Cancer Institute, National Institutes of Health, Bethesda, MD 20892, USA. [4] High-Dimension Data Analysis Group, Laboratory of Cancer Biology and Genetics, Center for Cancer Research, National Cancer Institute, National Institutes of Health, Bethesda, MD 20892, USA. [5] In Vitro Pathogenesis Section, Laboratory of Cancer Biology and Genetics, Center for Cancer Research, National Cancer Institute, National Institutes of Health, Bethesda, MD 20892, USA. [6] Cancer Genetics Section, Laboratory of Cancer Biology and Genetics, Center for Cancer Research, National Cancer Institute, National Institutes of Health, Bethesda, MD 20892, USA. [7] Transgenic Core, National Heart, Lung, and Blood Institute, National Institutes of Health, Bethesda, MD 20892, USA. Correspondence and requests for materials should be addressed to J.H. (email: huangj3@mail.nih.gov)

Dysregulations of translation and transcription are hallmarks of human cancer[1–5]. These two cellular processes are generally regulated by distinct machineries at different subcellular locations. In the nucleus, transcription factors regulate transcription to produce messenger RNAs (mRNAs), which are then transported into the cytoplasm for translation. One critical step of translation regulation is the initiation that involves the attachment of the translation pre-initiation complex (PIC) to activated mRNAs[6]. The initiation process can be executed in a cap-dependent and/or -independent manner[6]. In the cap-dependent mechanism, the eukaryotic initiation factor 4 F (eIF4F) complex, which comprises eIF4E, eIF4G, and a RNA helicase eIF4A, binds to the $m^7G$ cap at the 5′ untranslated region (5-′UTR) of mRNAs and unwinds the secondary structure to enhance PIC loading. In the cap-independent mechanism, other factors, such as eIF4B, directly stimulate the attachment of PIC to mRNAs and bypass the eIF4F complex. Although translation and transcription regulation have been extensively studied, it remains unclear whether they can be co-regulated in cancer and how their dysregulations lead to tumorigenesis.

The core binding factor subunit beta (CBFB) and its binding partner RUNX1 (also called AML1) regulate a diverse signaling pathways to maintain the homeostasis of a wide range of cell types and tissues[7,8]. The genetic alterations of CBFB and RUNX1 have been associated with many types of human disorders and cancers[9–11]. At the molecular level, RUNX1 and CBFB form a transcriptional complex. The well-accepted mechanism of action of the CBFB/RUNX1 complex is that RUNX1 is a sequence-specific DNA-binding transcription factor while CBFB has no DNA-binding activity but it heterodimers with RUNX1 in the nucleus and enhances the DNA-binding and transcriptional activity of RUNX1[9,10].

Although much is known about the roles of CBFB and RUNX1 in the hematopoietic system and blood cancer, our knowledge of their functions and regulatory mechanisms in other tissues and cancers is very limited. Recent genome-wide sequencing studies in breast tumors revealed that *CBFB* is highly mutated in human breast tumors, suggesting that CBFB plays critical roles in the etiology of breast tumor[12,13]. In this study, we set out to elucidate the function of CBFB in breast cancer and unexpectedly discover an unexpected role of CBFB in translation regulation. CBFB binds to and enhances the translation of *RUNX1* mRNA, which encodes the binding partner of CBFB. Using genome-wide approaches, we further show that CBFB binds and regulates the translation of hundreds of mRNAs. CBFB binds to mRNAs through hnRNPK and facilitate translation initiation by eIF4B. Our data support a model that CBFB has dual functions, regulating translation in the cytoplasm and transcription in the nucleus. Importantly, both the cytoplasmic and nuclear functions of CBFB are critical for suppressing breast cancer. We propose that breast cancer cells evade translation and transcription surveillance simultaneously by CBFB downregulation.

## Results

**Both CBFB and RUNX1 suppress breast cancer.** To study the function of CBFB in breast cancer, we generated CBFB knockout (KO) cell lines from MCF10A cells (Supplementary Fig. 1a), a non-tumorigenic human mammary epithelial cell line, using the clustered regularly-interspaced short palindromic repeats (CRISPR)-Cas9 technology. We then transfected CBFB_KO cells with plasmids expressing tumor-derived CBFB mutants. All these CBFB mutants had undetectable protein levels (Fig. 1a) while their mRNAs were comparable to that of CBFB wild type (WT) (Supplementary Fig. 1b), suggesting that these tumor-derived mutations destabilize CBFB and result in loss of function.

CBFB_KO MCF10A cells became transformed in vitro judged by the anchorage independent assay and formed tumors in immunocompromised NSG (NOD-scid, IL2R gamma[null]) mice (Fig. 1b, Supplementary Fig. 1c-d, and Supplementary Table 1). The transformation effect was reversed by CBFB overexpression, ruling out the off-target effect of guide RNAs of CBFB (Supplementary Fig. 1e, f and Supplementary Table 1). These data suggest that CBFB has a tumor suppressive function in breast cancer.

Interestingly, we observed a concurrent loss of RUNX1 protein upon CBFB deletion (Fig. 1c) and the loss was reversible by CBFB overexpression (Supplementary Fig. 1g). However, CBFB protein level was not affected by RUNX1 deletion (Fig. 1d). RUNX1 deletion phenocopied CBFB deletion and transformed MCF10A cells both in vitro and in vivo (Fig. 1e, Supplementary Fig. 1h, and Supplementary Table 1), which motivated us to study the regulation of RUNX1 by CBFB.

**The inter-regulation of CBFB and RUNX1.** In studying the subcellular localization of CBFB and RUNX1, we observed that CBFB and RUNX1 had different subcellular localizations; CBFB was mainly localized in the cytoplasm while RUNX1 was predominantly in the nucleus (Fig. 1f). This observation was consistent across multiple breast cell lines, including two non-tumorigenic breast cells (MCF10A and MCF12A), four luminal-type, and four triple-negative or basal-like breast cancer cells (Fig. 1f). We noted a significant reduction of RUNX1 in the luminal-type but not the triple negative type breast cancer cells compared to normal cells while the reduction of CBFB was modest. We confirmed the disparity of subcellular locations of CBFB and RUNX1 using immunocytochemistry (Fig. 1g) and immunohistochemistry (Supplementary Fig. 2a). The predominant cytoplasmic localization of CBFB was also observed in non-neoplastic human breast tissue (Supplementary Fig. 2b).

A small portion of CBFB is localized in the nucleus of breast cells (Fig. 1f). To test the effect of RUNX1 deletion on CBFB cytoplasmic and nuclear distribution, we performed fractionation in WT and RUNX1_KO cells and found that the nuclear population of CBFB was decreased in RUNX1_KO cells compared to WT cells (Fig. 1h). Thus, although RUNX1 does not affect the total steady-state levels of CBFB in the cell (Fig. 1d), it affects the levels of CBFB in the nucleus. This observation is consistent with a previous report showing that RUNX1 is required for the nuclear shuttling of CBFB[3].

We then turned our attention to the underlying mechanism of RUNX1 loss in CBFB deleted cells (Fig. 1c). A possibility is that CBFB binds to RUNX1 in the nucleus and prevents it from degradation. Blocking the proteasomal and lysosomal degradation pathways with MG132 and hydroxychloroquine (HCQ) did not rescue the loss of RUNX1 in CBFB_KO cells (Supplementary Fig. 3a, b), indicating that these two protein degradation pathways are not involved in RUNX1 loss. These results also imply that the regulation of RUNX1 protein by CBFB is independent of their interaction in the nucleus. To test this hypothesis, we took advantage of an observation from our previous study[14], wherein we studied CBFB and RUNX2, another member in the RUNX family. In that study, we found that an amino-terminal (N-terminal) fused FLAG tag prevented CBFB from binding to RUNX2 while a carboxyl-terminal (C-terminal) tag did not. Since CBFB is a common binding partner for RUNX1 and RUNX2, we reasoned that a N-terminal FLAG tag may block the interaction between CBFB and RUNX1. We performed co-IPs of RUNX1 and CBFB with either an N-terminal or C-terminal FLAG tag in MCF10A cells. Indeed, CBFB with a C-terminal FLAG tag interacted with RUNX1 while CBFB with a N-terminal tag did not (Fig. 1i). Further, both N-terminally and C-terminally

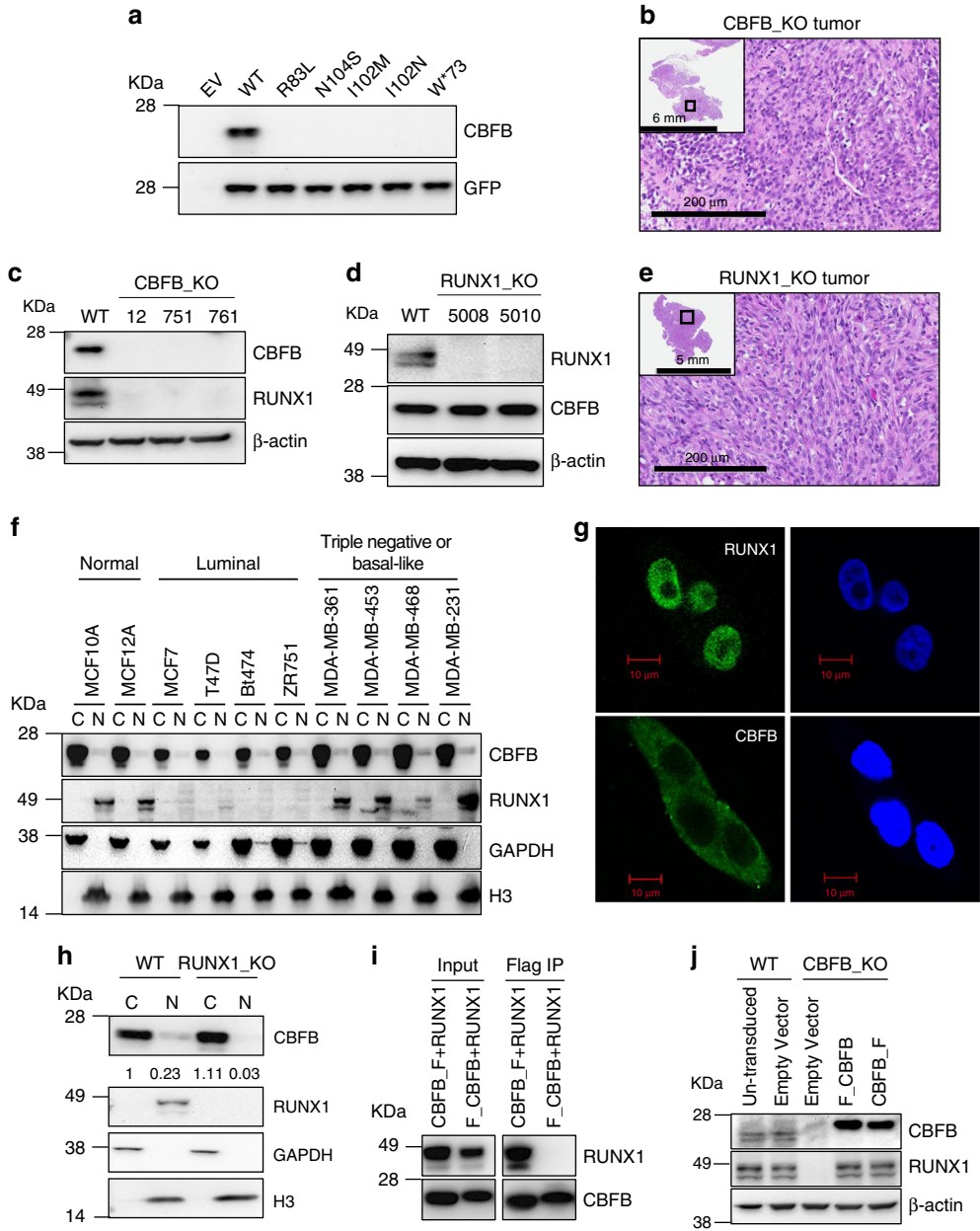

**Fig. 1** CBFB is a tumor suppressor and essential for maintaining RUNX1 protein levels. **a** IB showing expression of WT and CBFB mutants in CBFB_KO MCF10A cells. **b** Hematoxylin & eosin (H&E) staining of a representative xenograft tumor formed from subcutaneously injected CBFB_KO MCF10A cells. **c** IB showing the reduction of RUNX1 protein in CBFB_KO MCF10A cells. **d** IB showing RUNX1 deletion in MCF10A cells. **e** H&E staining of a representative tumor formed from RUNX1_KO MCF10A cells. **f** IB showing the subcellular localization of CBFB and RUNX1 in multiple breast cells. GAPDH, a marker for the cytoplasm (**c**); histone H3, a marker for the nucleus (N). **g** immunocytochemistry (ICC) showing the subcellular location of CBFB and RUNX1 in MCF10A cells. **h** IB showing the effect of RUNX1 deletion on the subcellular distribution of CBFB between the cytoplasm and nucleus. The numbers underneath the CBFB blot indicate the relative CBFB amounts quantified using ImageJ. **i** Co-immunoprecipitation (Co-IP) showing the interaction of RUNX1 with a N-terminal FLAG tag (F-CBFB) or a C-terminal tag (CBFB-F) in MCF10A cells. **j** IB showing the effect of overexpression of F-CBFB or CBFB-F on RUNX1 protein levels in CBFB_KO MCF10A cells

tagged CBFB were mainly localized in the cytoplasm (Supplementary Fig. 3c, d). Importantly, both N-terminally and C-terminally tagged CBFB rescued the expression of RUNX1 in CBFB_KO cells (Fig. 1j), demonstrating that the interaction between CBFB and RUNX1 is not required for CBFB to regulate RUNX1 protein level.

Results from real-time PCR and RNAseq ruled out the possibilities that CBFB regulates *RUNX1* transcription, *RUNX1* mRNA nuclear/cytoplasmic distribution, splicing, or degradation (Supplementary Fig. 3e–h). Together, these results suggest that

CBFB regulates RUNX1 protein through a previously unknown mechanism.

**CBFB binds to *RUNX1* mRNA via hnRNPK**. To search for the putative regulatory mechanism of RUNX1 by CBFB, we identified CBFB interacting proteins. To this end, we generated a stable CBFB KO MCF10A cell line expressing N-terminally FLAG tagged CBFB, performed FLAG IP, and subjected enriched bands for protein identification using mass spectrometry. Using this approach, we identified hnRNPK as a prominent binding partner

of CBFB (Fig. 2a, b and Supplementary Data 1). Using a stable cell line expressing C-terminally FLAG tagged CBFB, we also identified hnRNPK as a binding partner of CBFB (Supplementary Fig. 4a, b and Supplementary Data 1). We note that CBFB overexpression and deletion did not alter the levels of hnRNPK in MCF10A cells (Fig. 2b and Supplementary Fig. 4b, c), ruling out the possibility that the detection of the interaction between CBFB and hnRNPK is simply due to increased levels of hnRNPK by CBFB. The interaction between hnRNPK and CBFB was confirmed under endogenous condition (Supplementary Fig. 4d). We then mapped the interacting regions within CBFB and hnRNPK. Amino acid residues 1 to 141 within CBFB were critical for interacting with hnRNPK while residues 1 to 220 within hnRNPK were required for CBFB interaction (Supplementary Fig. 4e, f).

We surprisingly found that hnRNPK was localized in both cytoplasm and nucleus (Supplementary Fig. 4d, input in the cytoplasmic fraction). Given that hnRNPK is generally considered as a nuclear protein, we performed cell fractionation and immunofluorescence in multiple breast cell lines using three different antibodies that recognize different epitopes within hnRNPK. We detected a significant portion of hnRNPK in the cytoplasm of all the cell lines tested (Supplementary Fig. 5a, b). The cytoplasmic localization of hnRNPK was confirmed using normal human breast tissue (Supplementary Fig. 5c). Interestingly, only cytoplasmic hnRNPK interacted with CBFB (Supplementary Fig. 4d, see co-IP in the cytoplasmic and nuclear fractions).

hnRNPK is a multi-functional protein, which has been shown to play critical roles in transcription, pre-mRNA processing, and translation via binding to DNA and RNA[15–17]. Knockdown of hnRNPK using siRNA revealed that hnRNPK regulates RUNX1 protein levels (Fig. 2c). Like CBFB, hnRNPK did not affect the steady-state levels and the nucleus-to-cytoplasm distribution of RUNX1 mRNA (Supplementary Fig. 6a, b). We then explored the possibility that CBFB and hnRNPK regulate RUNX1 mRNA processing or translation. To test this hypothesis, we first examined the interaction of CBFB and hnRNPK with RUNX1 mRNA by performing RIP (RNA immunoprecipitation). Both CBFB or hnRNPK antibodies efficiently pulled down RUNX1 mRNA but not GAPDH and 28 S ribosomal RNAs (Fig. 2d, e). CBFB deletion decreased the recruitment of hnRNPK to RUNX1 mRNA (Fig. 2e) and vice versa (Supplementary Fig. 6c), suggesting that they cooperatively bind to the RNA. The binding of CBFB and hnRNPK to RUNX1 mRNA was mainly in the cytoplasm (Supplementary Fig. 6d). We also examined the binding of two poly-C binding hnRNPs, hnRNPL and hnRNPE2, to RUNX1 mRNA. hnRNPL did not bind to RUNX1 mRNA while hnRNPE2 did (Supplementary Fig. 6d, e). However, CBFB neither affected the binding of hnRNPE2 to RUNX1 mRNA (Supplementary Fig. 6e) nor bound to hnRNPE2 (Supplementary Fig. 6f). Binding of hnRNPK and CBFB to RUNX1 mRNA was also detected in several other breast cancer cell lines (Supplementary Fig. 6g–j), indicating that binding of hnRNPK and CBFB to RUNX1 mRNA is a general mechanism.

To determine the regions within RUNX1 mRNA that are bound by CBFB and hnRNPK, we performed RNA pulldown assays (RPA) using biotin labeled RNA fragments of RUNX1 mRNA and MCF10A cell lysate. A 1141-nucleotide (nt) fragment (called F3) within the 3′ untranslated region (UTR) of RUNX1 mRNA interacted with CBFB and hnRNPK (Fig. 2f). We further narrowed down the CBFB/hnRNPK binding region to a 226-nt truncated region (called T14) (Supplementary Fig. 6k). When binding to RNA, hnRNPK has previously been shown to be a poly cytosine (poly-C) binding protein[16]. Within the T14 region, there are three poly-C tracts. We then carried out site-directed mutagenesis and changed Cs to adenosines (As) or thymidines

(Ts) within these poly-C tracts (Supplementary Fig. 6l). The first poly-C tract played a major role for the binding of CBFB and hnRNPK since its alteration completely abolished the binding of these two proteins to RUNX1 mRNA (Supplementary Fig. 6l). The second poly-C tract was also involved in the binding but played a smaller role than the first one, and the third one did not mediate the binding. Re-analysis of two public eCLIP datasets of hnRNPK[18] revealed a hnRNPK binding site at the 3′ UTR of RUNX1 mRNA containing the first and second poly-C tracts (Fig. 2g).

To test whether CBFB and hnRNPK directly bind to RUNX1 mRNA, we performed RPA using recombinant CBFB and hnRNPK and biotin labeled RNA. CBFB did not directly interact with RUNX1 mRNA while hnRNPK did (Fig. 2h, i). Recombinant CBFB greatly increased the binding of hnRNPK to RUNX1 mRNA (Fig. 2i). Thus, hnRNPK directly binds to the 3′ UTR of RUNX1 mRNA. CBFB interacts with hnRNPK and enhances its binding to RUNX1 mRNA.

**CBFB binds to hundreds of transcripts through hnRNPK.** After establishing that CBFB and hnRNPK bind to RUNX1 mRNA, we asked whether CBFB binds to other transcripts. To this end, we performed RIP followed by deep sequencing (RIPseq) and identified 837 CBFB-bound transcripts (fold enrichment > 4) (Fig. 3a) and 1752 hnRNPK-bound transcripts (Fig. 3b). Among the 837 CBFB-bound transcripts, 755 (90%) were also bound by hnRNPK (Fig. 3c). We selected 16 transcripts for further validation because they are relatively well-known and thus antibodies are available for their encoded proteins. We detected binding of CBFB and hnRNPK to all these transcripts (Fig. 3d). Examination of the public eCLIP dataset of hnRNPK[18] revealed that 530 (70%) out of the 755 common binding transcripts of CBFB and hnRNPK have at least one eCLIP site of hnRNPK. Using the GLAM2 algorithm[19], we identified one gapped motif containing poly-C (Fig. 3e). This gapped motif was represented in 86% of the hnRNPK-bound transcripts. These data strongly suggest that CBFB interacts with these transcripts through hnRNPK, which directly binds to these RNAs.

**CBFB and hnRNPK have direct roles in translation regulation.** Because one of the functions of hnRNPK is translation regulation[20] and we demonstrated CBFB and hnRNPK bind to hundreds of mRNAs, we hypothesized that CBFB regulates protein translation. To assess the direct roles of CBFB and hnRNPK in RUNX1 translation, we used an in vitro translation assay (Fig. 4a). RNA encoded by RUNX1 cDNA alone had residual translation activity (Fig. 4b), suggesting that other elements are required. The T14 fragment of the 3′ UTR of RUNX1 mRNA, to which CBFB and hnRNPK bind, greatly enhanced the translation activity (Fig. 4b). Importantly, disrupting the binding of CBFB and hnRNPK (by Mu1, Mu2, or Mut1 + 2) completely abolished the activity of the T14 fragment in enhancing RUNX1 translation, suggesting that the binding of CBFB and hnRNPK to RUNX1 mRNA is critical for RUNX1 translation. Additional recombinant hnRNPK alone did not induce RUNX1 translation (Supplementary Fig. 7a), suggesting that the amount of hnRNPK is saturated in HeLa cell lysate or that its effect is limited by the availability of other factors, such as CBFB. In contrast to recombinant hnRNPK, recombinant CBFB enhanced RUNX1 translation (Fig. 4c). Interestingly, although recombinant hnRNPK alone did not enhance RUNX1 translation, it greatly potentiated the effect of recombinant CBFB on RUNX1 translation (Supplementary Fig. 7b), indicating that CBFB and hnRNPK cooperatively enhance RUNX1 translation and CBFB is a rate-limiting factor. To further validate the roles CBFB and hnRNPK in RUNX1

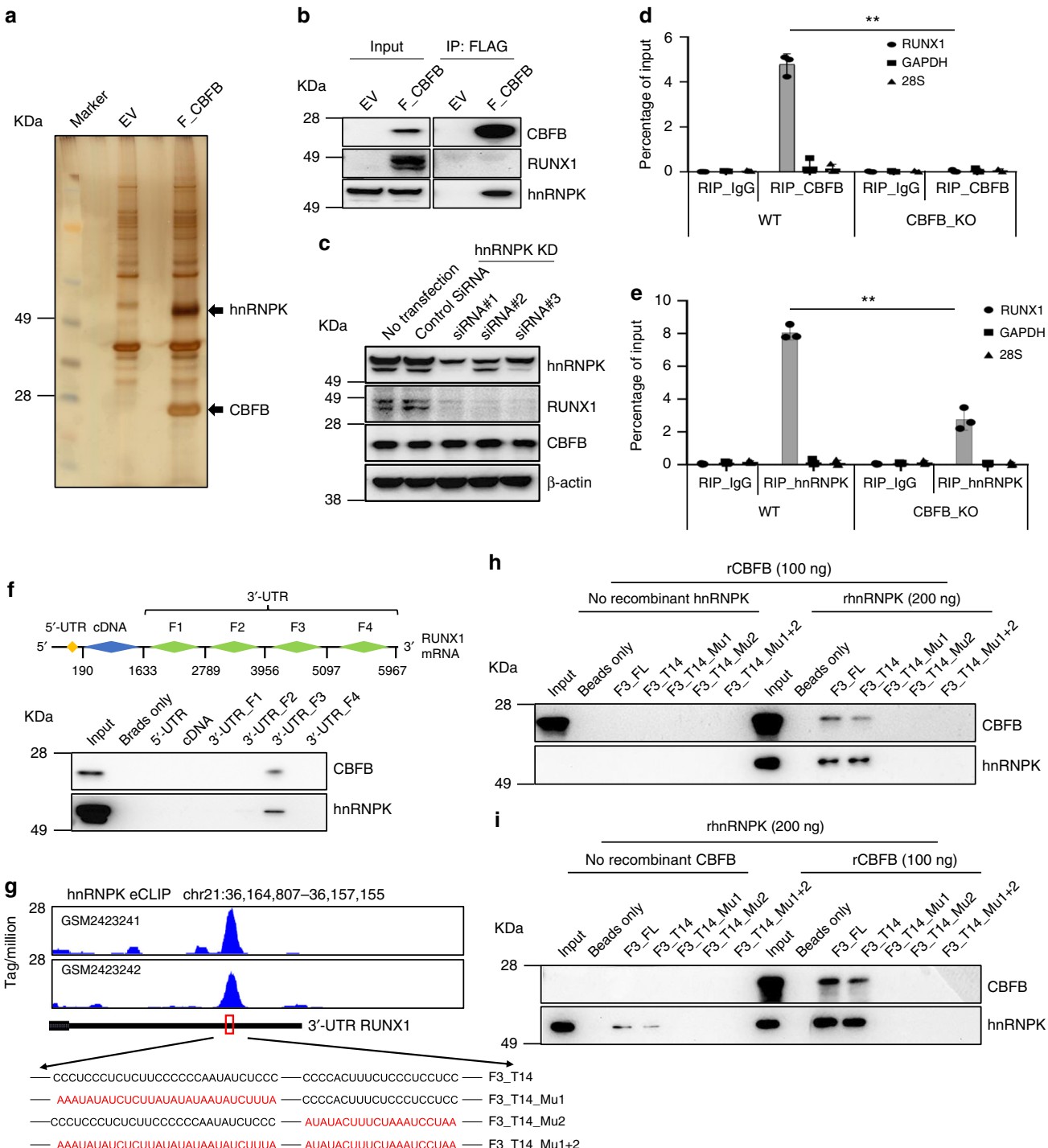

**Fig. 2** CBFB binds to *RUNX1* mRNA via hnRNPK. **a** Silver staining of FLAG pulldown using CBFB KO MCF10A cells expressing empty vector and N-terminally FLAG tagged CBFB. **b** IB validation of CBFB and hnRNPK interaction. **c** IB showing the effect of hnRNPK knockdown on RUNX1 protein. **d** RNA immunoprecipitation (RIP) with CBFB antibody in WT and CBFB_KO MCF10A cells. Error bars are SEM, $n = 3$ (biological); two asterisks, $p$ value < 0.01. (RIP of CBFB in WT vs. in CBFB KO cells, two-tailed $t$ test). **e** RIP with hnRNPK antibody. Error bars are SEM, $n = 3$ (biological); two asterisks, $p$ value < 0.01 (RIP of hnRNPK in WT vs. CBFB KO cells, two-tailed $t$ test). **f** RNA pulldown assays (RPA) determining the hnRNPK-bound region within *RUNX1* mRNA. F1-4, fragment 1 to 4 of 3′ UTR of *RUNX1* mRNA. See Methods for details. Numbers indicate the nucleotide positions. **g** Re-analyses of two public eCLIP datasets (GSM2423241 and GSM2423242) of hnRNPK on the RUNX1 locus. Nucleotide sequences of two poly-C tracts within the binding site of hnRNPK were shown. F3, fragment 3 of 3′-UTR; T14, truncation 14 of 3′-UTR F3; Mu1, mutation 1; Mu2, mutation 2. **h** RPA using recombinant CBFB in the absence or presence of recombinant hnRPNK. **i** RPA using recombinant hnRNPK in the absence or presence of recombinant CBFB

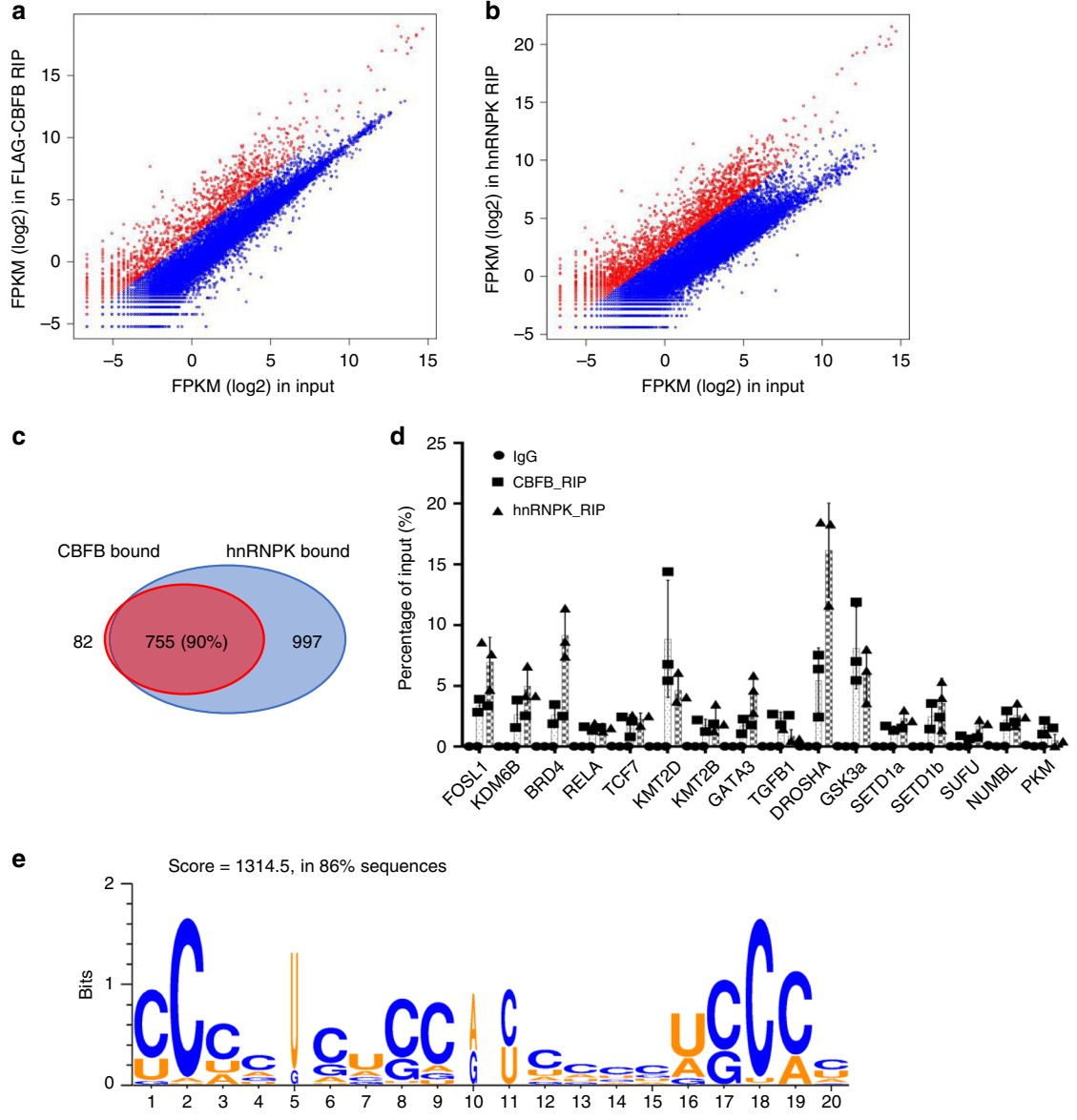

**Fig. 3** Genome-wide binding of CBFB and hnRNPK to mRNAs. RIPseq of FLAG-CBFB (**a**) and hnRNPK (**b**) in MCF10A cells. Shown are mean adjusted FPKM $+1$ (log2). Red dots show the transcripts that are enriched in FLAG-CBFB RIP more than 4-fold. The rest transcripts are shown as blue dots. **c** A Venn diagram showing transcripts bound by both CBFB and hnRNPK. **d** Validation of binding of CBFB and hnRNPK to 16 selected transcripts using RIP followed by real-time PCR. Error bars are SEM, $n = 3$ (biological). **e** The top-ranked gapped motif, which was represented in 86% of the hnRNPK binding sites occurred in hnRNPK-bound transcripts

translation, we generated CBFB_KO and hnRNPK_KD 293 T cells (Supplementary Fig. 7c, d). The reason for choosing 293 T is two-fold. First, 293 T cells do not express detectable RUNX1; therefore, the result of in vitro translation assay was not affected by endogenous RUNX1 protein. Second, CBFB and hnRNPK protein levels and subcellular localization were comparable in Hela and 293 T cells (Supplementary Fig. 7e, f). We prepared CBFB_WT, CBFB_KO, hnRNPK_KD 293 T lysates and used them in the in vitro translation assay. RUNX1 translation was greatly reduced in the translation reaction using CBFB_KO or hnRNPK_KD lysate compared to controls (Fig. 4d, e). Together, these data establish a direct role of CBFB and hnRNPK in RUNX1 translation.

We then examined whether CBFB and hnRNPK regulate the translation of other transcripts. We tested the expression of 10 proteins encoded by mRNAs bound by CBFB and hnRNPK. Nine out of the ten proteins had reduction in CBFB_KO or

hnRNPK_KD cells compared to WT cells (Fig. 4f, g). Importantly, the mRNA levels of these genes did not change (Supplementary Fig. 7g, h), suggesting that CBFB and hnRNPK regulate their translation. Thus, CBFB and hnRNPK have a wide role in translation regulation.

**CBFB regulates translation initiation.** To study the mechanism of translation regulation by CBFB, we performed polysome profiling and measured *RUNX1* mRNA in the free mRNA, mono-, and polyribosome fractions[21] (Fig. 5a and Supplementary Fig. 7i). CBFB deletion greatly shifted the distribution of *RUNX1* mRNA in the free mRNA, mono-, and pooled polyribosomal fractions (Fig. 5b). The population of *RUNX1* mRNA in the free mRNA fraction greatly increased, suggesting that CBFB is essential for translation initiation of the RNA. The ratios of *RUNX1* mRNA in pooled poly- to monoribosomal fractions were similar in WT and

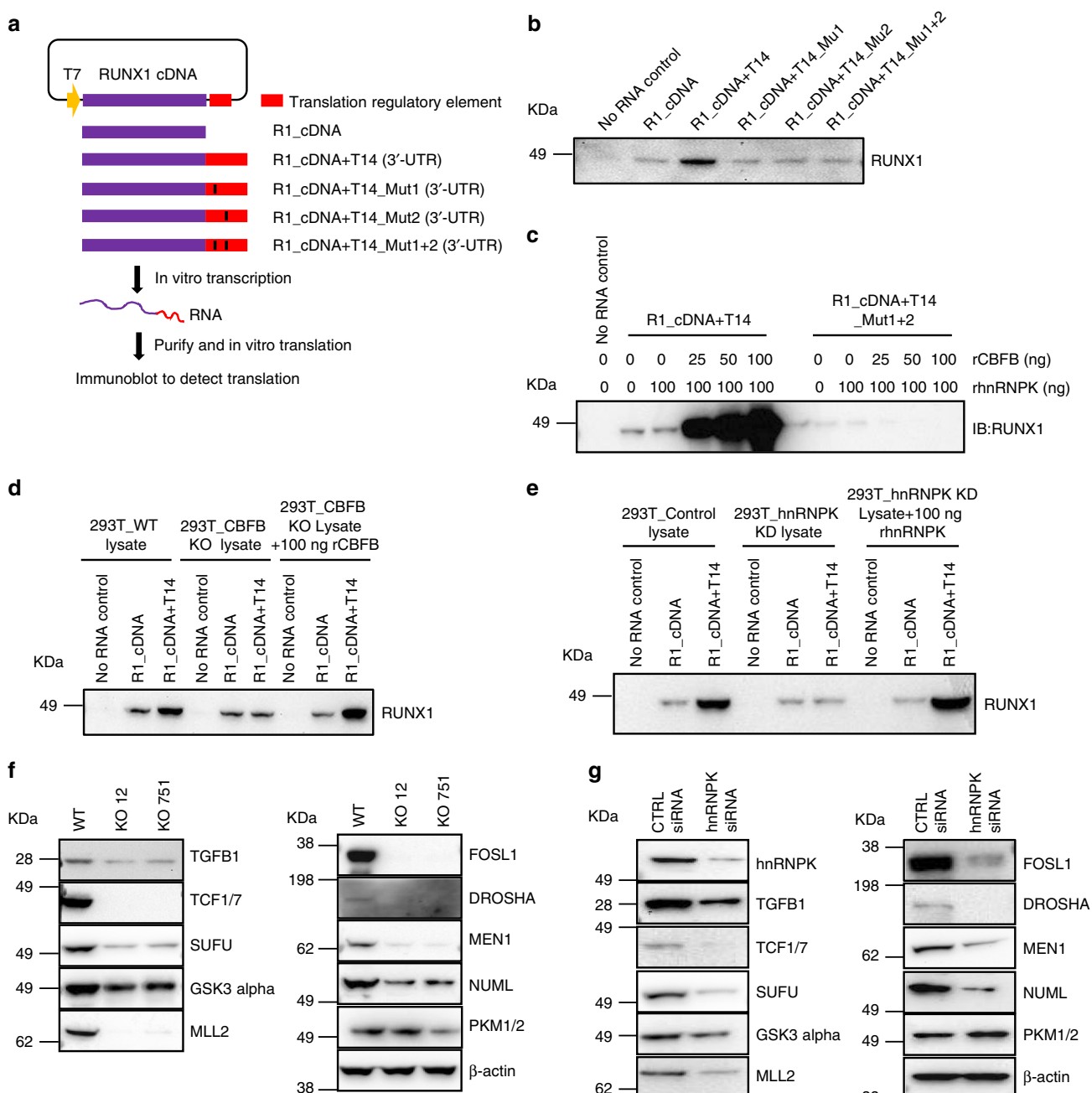

**Fig. 4** CBFB and hnRNPK regulate translation. **a** Flow chart showing the in vitro translation assay to test the effect of a translation element (RNA) and a protein. In vitro transcribed RNA was used in the in vitro translation assays using HeLa cells lysate. **b** In vitro translation assay using the 1-Step Human coupled IVT kit showing the effect of CBFB and hnRNPK binding sites (T14) on RUNX1 translation. **c** In vitro translation assays using the 1-Step Human coupled IVT showing effect of recombinant CBFB (rCBFB) and hnRNPK (rhnRNPK) on RUNX1 translation. **d** In vitro translation assay using WT and CBFB_KO HEK 293 T cells lysate. See "Methods" section for details of preparation of 293 T cell lysate. **e** In vitro translation assay. Un-transfected (control) and hnRNPK knockdown (KD) HEK 293T cells lysate replaced the HeLa cell lysate in the 1-Step Human coupled IVT kit with or without the supplement of rhnRNPK. **f** IB showing the effect of CBFB deletion on the proteins encoded by bound transcripts. **g** IB showing the effect of hnRNPK KD on the proteins encoded by bound transcripts

CBFB_KO cells, indicating that CBFB is not involved in translation elongation of *RUNX1* (see Fig. 5c numbers). In contrast, the distribution of *GAPDH* mRNA in the free, mono-, and pooled polyribosomal fractions did not change upon CBFB deletion (Fig. 5d, e). This result is consistent with the observation that *GAPDH* mRNA is not bound by CBFB (Fig. 2d). In addition, we also examined whether CBFB deletion affects the distribution of other transcripts bound by CBFB and hnRNPK in the free, mono-, and pooled polyribosomal fractions. CBFB deletion increased

the percentage of all the mRNAs except *PKM* in the free fraction (Fig. 5f). These data are consistent with the observation that the levels of proteins encoded by these mRNAs except *PKM* decreased in CBFB_KO and hnRNPK_KD cells (Fig. 4f, g). To assess the role of CBFB in translation regulation at the genome-wide level, we performed RNAseq of ribosomal fractions of WT and CBFB_KO cells (Fig. 5g). Globally, 27% of mRNAs had decreased translation, judged by a 50% decrease of the ratio (pooled polyribosomal to free fraction) when CBFB was deleted.

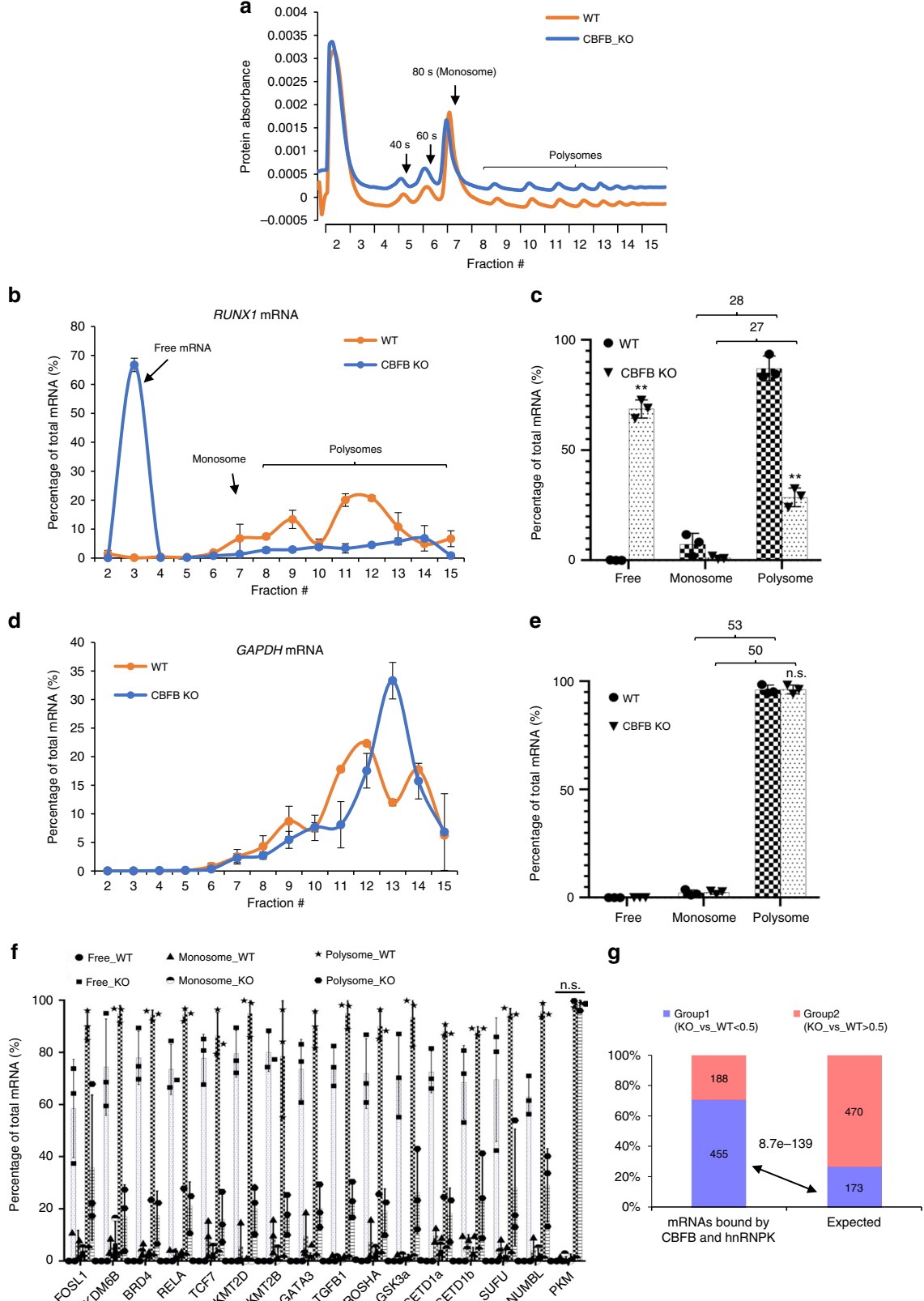

If only CBFB and hnRNPK-bound mRNAs were considered, 70% had decreased translation, indicating that CBFB and hnRNPK directly regulate translation initiation of majority of these mRNAs (Chi-square test, p = 8.7e-139). Therefore, the role of CBFB in translation initiation regulation is widespread.

**CBFB regulates translation initiation through eIF4B**. To further investigate the underlying mechanism of CBFB-regulated translation initiation, we searched for translation related factors in identified CBFB interacting proteins (Fig. 2a) and detected eukaryotic initiation factor 4B (eIF4B) as a potential CBFB

**Fig. 5** CBFB regulates translation initiation. **a** Protein absorbance of each fraction of ribosome fractionation of WT and CBFB_KO MCF10A cells. **b** Real-time PCR measuring the abundance of *RUNX1* mRNA in each fraction. Shown are the percentages of total *RUNX1* mRNA in each fraction. Error bars are SEM, $n = 3$ (biological repeats). **c** Percentage of total *RUNX1* mRNA in the monosome and pooled polyribosomal fractions. Numbers on top of brackets are the ratios of *RUNX1* mRNA amount in pooled poly- to monoribosomal fractions. Error bars are SEM, $n = 3$ (biological), two asterisks, *p* value < 0.01 (CBFB KO vs. WT comparisons in the free and polysome fractions, two-tailed *t* test). **d** Real-time PCR measuring the percentage of total *GAPDH* mRNA in each fraction. Error bars are SEM, $n = 3$ (biological repeats). **e** Percentage of total *RUNX1* mRNA in the mono- and pooled polyribosome fractions. Numbers are the ratios of *GAPDH* mRNA amount in pooled poly- to monoribosomal fractions. Error bars are SEM, $n = 3$ (biological); n.s., *p* value > 0.05 (CBFB KO vs. WT comparisons in the free and polysome fractions, two-tailed *t* test). **f** Real-time PCR measuring percentage of each transcript in the free, mono- and polyribosomal fractions in WT and CBFB_KO MCF10A cells. Error bars are SEM, $n = 3$ (biological); n.s., *p* value > 0.05. **g** Bar chart showing the percentage of group 1 and group2 in CBFB/hnRNPK-bound mRNAs. Group 1, ratio of polyribosomal vs. free fraction decreased more than half in CBFB_KO compared to WT cells; Group 2, the rest of mRNAs. The expected percentage is based on the global mRNAs. Chi-square test was used to calculate the *p* value

binding protein (Supplementary Data 1). Since only enriched bands were subject to mass spectrometry analysis, our approach only identified potential binding partners of CBFB with high stoichiometric values. Therefore, we also screened several additional translation initiation factors and found that eIF4B was the major translation initiation factor binding to CBFB (Fig. 6a). We also detected the interaction between eIF4B and CBFB under endogenous condition (Fig. 6b). The interaction was not RNA-mediated since RNase A was added into the co-IP lysate (Fig. 6b). eIF4B stimulates translation initiation through several non-exclusive mechanisms. It has been shown to stimulate the RNA helicase activity of eIF4A, bind to eIF3a to bridge the interaction of PIC and eIF4F, or directly enhance PIC attachment to mRNAs via its RNA binding domain[6,22,23]. We performed the in vitro translation assay using RNA containing an m[7]G cap (cap dependent) or an encephalomyocarditis virus (EMCV) internal ribosome entry site (IRES) element (cap independent) (Fig. 6c). In the presence of rCBFB, recombinant eIF4B (reIF4B) stimulated translation of RUNX1 in both cap dependent and independent manner. Interestingly, either reIF4B or rhnRNPK alone did not enhance RUNX1 translation in the absence of rCBFB, suggesting that rCBFB is the limiting factor. CBFB deletion completely abolished the binding of eIF4B to CBFB-bound transcripts (Fig. 6d), consistent with the observation that CBFB is involved in translation initiation (Fig. 5). Together, our data support a model that CBFB regulates translation initiation through the initiation factor eIF4B (Fig. 6e).

**The nuclear CBFB/RUNX1 complex represses NOTCH3.** After establishing the cytoplasmic role of CBFB in translation regulation and breast tumor suppression, we aim to assess whether its nuclear function in transcription regulation is also involved in the tumor suppressive function. To this end, we performed RNA-seq analysis in RUNX1_WT, RUNX1_KO, CBFB_WT, and CBFB_KO in MCF10A cells (Fig. 7a). There were 212 genes that were commonly regulated by CBFB and RUNX1. Among these genes, 138 genes were induced and 74 were repressed by both CBFB and RUNX1. A gene signature including the top 10 genes (see Supplementary Methods for gene names in the signature) induced by CBFB and RUNX1 significantly predicted the Disease-Free Survival Probability of breast cancer patients from TCGA (The Cancer Genome Altas)[24] and the Disease Specific Survival of breast cancer patients from METABRIC[25] (Fig. 7b and Supplementary Fig. 8a). Pathway analysis using genes regulated by CBFB and RUNX1 revealed that the NOTCH signaling pathway is regulated by the nuclear CBFB/RUNX1 complex (Supplementary Table 2). Notably, NOTCH3 was highly upregulated in CBFB_KO and RUNX1_KO MCF10A cells (Fig. 7c). Restoration of CBFB and RUNX1 expression in their corresponding knockout cells rescued the upregulation of NOTCH3 (Supplementary Fig. 8b, c). In addition, hnRNPK knockdown increased NOTCH3 protein levels (Supplementary Fig. 8d),

consistent with the result that hnRNPK positively regulates RUNX1 (Fig. 2c). These results suggest that the nuclear CBFB/RUNX1 complex suppresses breast cancer by repressing the transcription of NOTCH3. It is worth noting that repression of the Notch signaling pathway by the RUNX transcription factor (Lz in *Drosophila*) was reported in *Drosophila* hematopoiesis[26], suggesting that the regulation of NOTCH by RUNX1 may be evolutionarily conserved. To investigate whether RUNX1 directly regulates NOTCH3, we performed ChIPseq of RUNX1. Probably due to the low affinity of our antibody or other un-identified reasons, we did not detect endogenous binding of RUNX1 on the NOTCH3 locus in MCF10A. However, using a MCF10A cell line that inducibly expresses RUNX1, we identified a binding site in the NOTCH3 locus, suggesting that RUNX1 may directly regulate NOTCH3 (Fig. 7d). To test whether NOTCH3 upregulation underlies the transformation ability of CBFB and RUNX1 deletion, we depleted NOTCH3 using CRISPR-Cas9 in CBFB_KO and RUNX1_KO MCF10A cells (Supplementary Fig. 8e). NOTCH3 deletion completely abolished the anchorage independent growth abilities (Fig. 7e and Supplementary Fig. 8f) and tumorigenicity (Supplementary Table 3) of CBFB_KO and RUNX1_KO cells. In addition, overexpression of NOTCH3 is sufficient for transforming MCF10A cells (Supplementary Fig. 8g-j and Supplementary Table 4). In summary, NOTCH3 repression contributes to the tumor suppressive function of the nuclear CBFB/RUNX1 complex. Thus, the canonical role of CBFB in transcription regulation is also critical for breast cancer suppression.

In luminal-type breast cancer cell lines, the levels of CBFB were lower compared to MCF10A, MCF12A, and triple negative/basal type breast cancer cell lines (Supplementary Fig. 8k). The CBFB cDNA was cloned and sequenced in these cell lines. No mutation was found, implying that in addition to gene mutations, other unknown mechanisms could lead to the downregulation of CBFB protein in breast cancer. Using human breast tumor microarray, we found that CBFB and RUNX1 levels were significantly lower in breast cancer tissues compared to normal breast tissues while NOTCH3 levels were the opposite (Fig. 7f and Supplementary Fig. 9). These results are consistent with our observation that CBFB/RUNX1 represses NOTCH3.

We then examined whether overexpression of CBFB in breast cancer cells decreases the tumorigenic ability of these cells. Overexpression of CBFB in MCF7 and BT474 cells reduced NOTCH3 and upregulated RUNX1 (Fig. 7g). Furthermore, overexpression of CBFB significantly reduced the tumorigenicity of MCF7 cells that were orthotopically injected into the mammary fat pad of athymic nu/nu mice (Nude mice) (Fig. 7h, i). These results further support the notion that the nuclear CBFB/RUNX1 complex suppresses breast cancer by repressing NOTCH3 and provide a proof-of-concept that upregulation of CBFB in breast cancer cells reduces tumor growth.

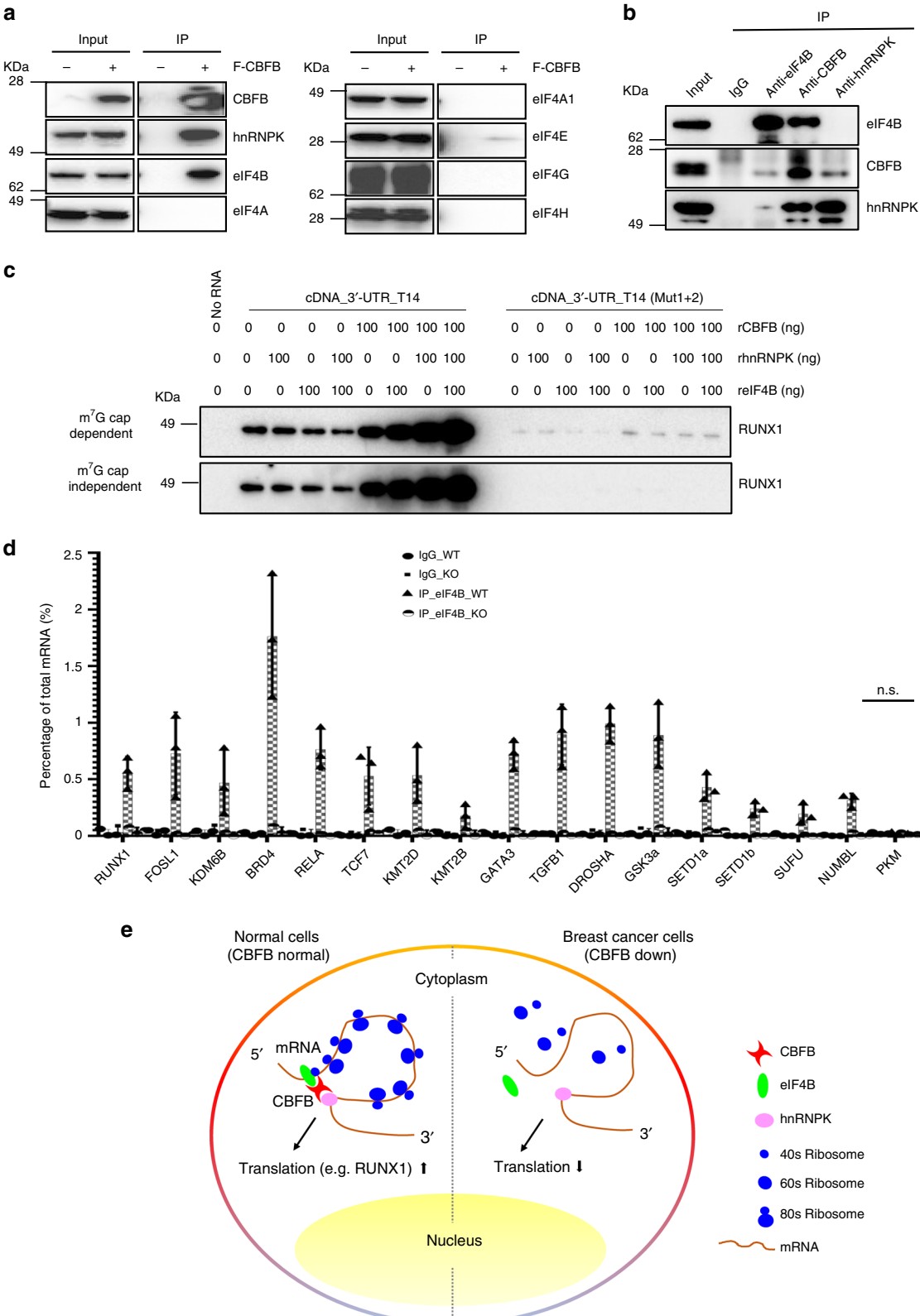

**Fig. 6** CBFB interacts with and facilitates mRNA binding of eIF4B. **a** FLAG IP followed by IB showing the interaction of translation initiation factors with CBFB. 10 μg/ml RNase A was added in the co-IP lysate. **b** Endogenous co-IP followed by IB showing interaction of CBFB, hnRNPK, and eIF4B. 10 μg/ml RNase A was added in the co-IP lysate. **c** In vitro translation assays showing the effect of rCBFB, rhnRNPK, and reIF4B on RUNX1 translation in a cap-dependent or -independent manner. **d** RIP followed by real-time PCR to show the effect of CBFB deletion on the binding of eIF4B on CBFB-bound transcripts. Error bars are SEM, $n = 3$ (biological); n.s., $p$ value > 0.05 (RIP of eIF4B in WT vs. CBFB KO cells). **e** A model showing the role of CBFB in translation regulation through hnRNPK and eIF4B

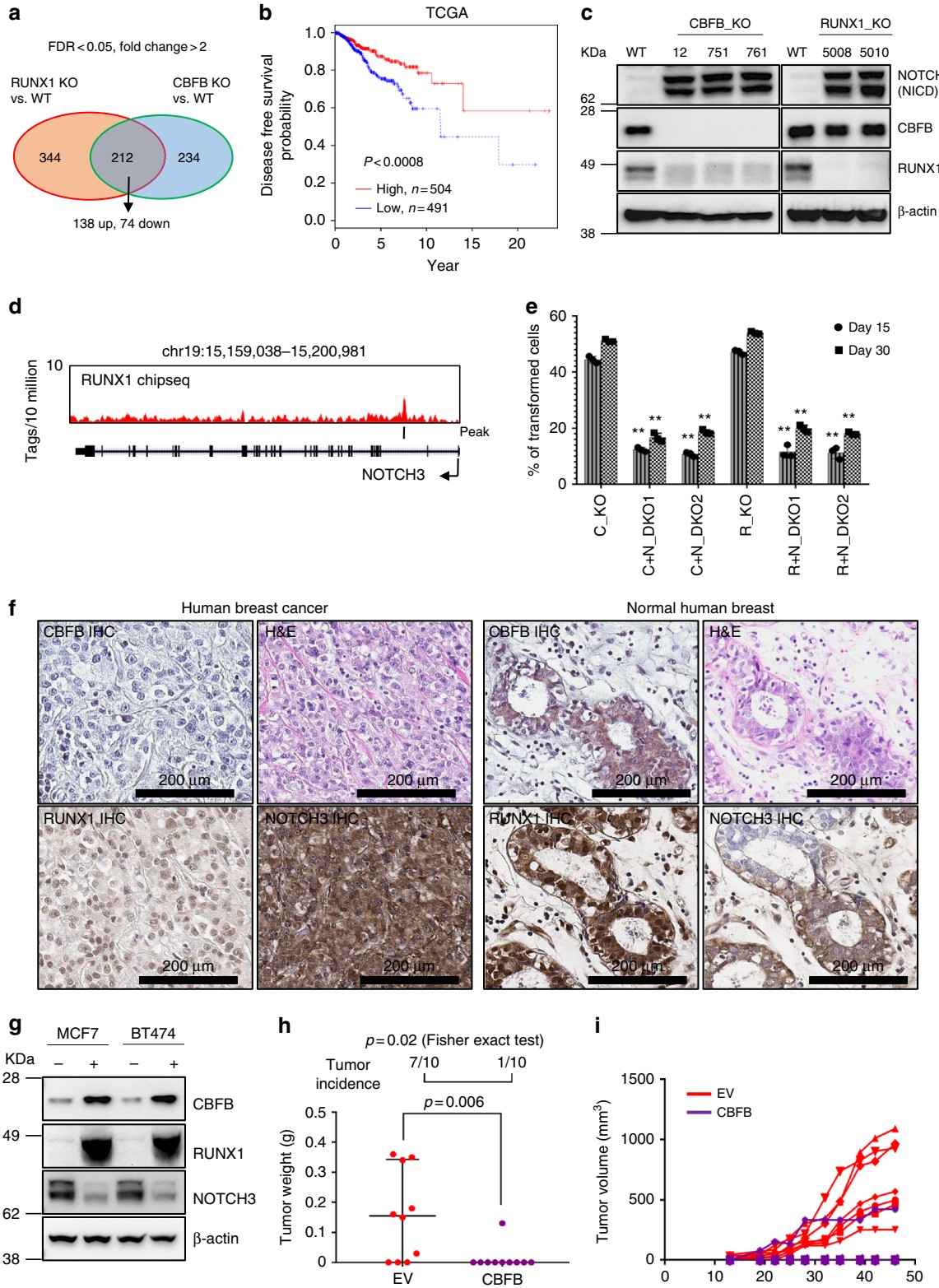

**Fig. 7** The transcriptional function of CBFB is critical for breast tumor suppression. **a** A Venn diagram showing genes (identified by RNAseq) co-regulated by CBFB and RUNX1. **b** Kaplan-Meier curve showing 10-gene signature upregulated by CBFB and RUNX1 predicts disease-free survival of breast cancer patients. **c** IB showing the upregulation of Notch intracellular domain (NICD) of NOTCH3 in CBFB_KO and RUNX1_KO cells. **d** ChIPseq of RUNX1 showing the binding of RUNX1 on the NOTCH3 locus. **e** Anchorage independent growth assays of double knockout (DKO) of CBFB (C) and NOTCH3 (N) or DKO of RUNX1(R) and NOTCH3(N). n = 3 (biological); two asterisks, p value < 0.01 (single KO vs. double KO comparisons, two-tailed t test). **f** Representative IHC images showing the protein levels of CBFB, RUNX1, and NOTCH3 in human breast tumors and normal tissues. **g** IB showing the effect of exogenous expression of CBFB on RUNX1 and NOTCH3 expression in MCF7 and BT474 cells. **h** Weight and (**i**) volume of orthotopically transplanted tumors from MCF7 cells that were transduced with an empty vector (EV) or a lentiviral vector expressing CBFB (CBFB). Tumor incidence is shown on the top

## Discussion

Our data reveal that cytoplasmic CBFB has a noncanonical role in translation regulation. The cytoplasmic localization of CBFB has been previously reported in mouse fibroblast cells[27,28]. However, it remains unknown whether cytoplasmic CBFB has a function. Since CBFB has no nuclear localization signal, the generally accepted view is that CBFB is made in cytoplasm and then shuttled into the nucleus to regulate transcription. Our data suggest that the cytoplasm is not merely a "parking place" for CBFB. Instead, it is the place where CBFB actively regulates translation of hundreds of transcriptions including *RUNX1* mRNA, which encodes the binding partner of CBFB. It has been proposed that CBFB is shuttled into the nucleus by the RUNX proteins[28]. Indeed, in this study, we also observed that RUNX1 is essential for maintaining the nuclear levels of CBFB (Fig. 1h). Thus, a possible scenario is that cytoplasmic CBFB enhances the translation of its own shuttle protein (RUNX1) to enter the nucleus and perform its nuclear functions, such as transcription regulation.

What is the relative importance of the role of CBFB in translation regulation compared to its canonical role in transcription regulation? In RIPseq, using 4-fold enrichment as a cutoff, CBFB bound to 837 out of 12388 transcripts (~7% of the transcriptome); At a cutoff of 2, it bound to 1606 transcripts (~14%) (Fig. 3a). Polyribosomal fractionation followed by RNAseq showed that about 500 mRNAs were potentially regulated by CBFB at the translation level (Fig. 5g). In contrast, CBFB deletion affected the transcription of 446 genes with a fold-change at 2 (Fig. 7a). Thus, the number of transcripts potentially regulated by CBFB at the translation level is comparable to that of transcripts regulated by CBFB at the transcription level, at least in our model. Although it is difficult to directly compare the relatively importance of the role of CBFB in translation regulation to its role in transcription regulation, our results show that the role of CBFB in translation regulation is widespread and worth further studying.

Our data do not support the notion that CBFB is a general translation factor. Instead, we propose that CBFB selectively regulates the translation of a subset of transcripts. An immediate question is how CBFB selects a transcript for translation regulation. Because CBFB does not directly bind to RNA, eIF4B and/or hnRNPK may generate the specificity in RNA selection, as both have RNA binding domains. Our data suggest that hnRNPK plays a major role in selecting mRNAs (Fig. 3) and CBFB may act as a bridging factor for eIF4B and hnRNPK to form an mRNA closed loop (Fig. 6e), which is proposed to promote translation[6].

Recently, CBFB and RUNX1 mutations have been identified as drivers in a variety of cancer types, including breast, ovarian, and prostate cancer[29,30]. Several lines of evidence support the notion that CBFB and RUNX1 are tumor suppressors in breast cancer. First, breast tumor-derived mutations of CBFB are loss of function (Fig. 1a). Second, deletion of CBFB or RUNX1 transforms MCF10A cells (Fig. 1b–e). Finally, downregulation of CBFB and RUNX1 are found in several breast cancer cell lines and human breast cancer tissues (Supplementary Fig. 8 and 9). Restoration of CBFB in these cell lines decreased the tumorigenicity (Fig. 7g–i). However, the CBFB gene is not mutated in these breast cancer cell lines, suggesting that downregulation of CBFB could be through either gene mutation, as shown by whole-genome sequencing studies, or other mechanisms. Therefore, the frequency of CBFB downregulation in breast tumor may be underestimated by genome sequencing studies.

Our data show that both the cytoplasmic and nuclear functions of CBFB are important for its breast tumor suppressive function. In the cytoplasm, CBFB is essential for the translation of many mRNAs including *RUNX1* mRNA. RUNX1 shuttles CBFB into the nucleus and forms the CBFB/RUNX1 transcriptional complex to regulate the transcription of many genes, one of which is NOTCH3. Thus, CBFB regulates gene expression at both translation level and transcription level, although genes translationally regulated by CBFB may be different from those regulated by CBFB transcriptionally. Since dysregulations of translation and transcription are hallmarks of tumorigenesis, breast tumor cells may overcome the barriers of translation and transcription surveillance simultaneously by CBFB downregulation.

It is worth noting that the tumor suppressive function of CBFB might be tumor subtype specific. In luminal-type breast cancer cell lines (MCF7, T47D, BT474, and ZR751), RUNX1 levels were almost undetectable and CBFB levels were lower than in triple negative and basal-like types of breast cancer cell lines (Fig. 1f). All these luminal-type breast cancer cell lines express estrogen receptor. Interestingly, a genome-wide sequencing study has suggested a potential association of CBFB mutations with estrogen-receptor-positive breast tumors although the observation was not statistically significant due to the small sample size[12]. Thus, more detailed studies are needed to establish whether CBFB function is linked to subtype of breast cancer. Nonetheless, CBFB has a tumor suppressive function in breast tumor and possible in other types of tumors, such as ovarian and prostate tumors. Because transcription activity is difficult to target, the role of CBFB in translation regulation may be therapeutically exploited for cancer treatment[2].

## Methods

**Reagents**. Recombinant human insulin (Cat# 910077 C), hydrocortisone (Cat# H0888g) and cholera toxin (Cat# C8052-5MG) were purchased from Sigma. Human Epidermal growth factor (EGF) was purchased from PeproTech (Cat# AF100-15). Proteosome inhibitor MG132 was procured from CalBiochem (Cat# 474790). Autophagy inhibitor hydroxychloroquine sulfate (HCQ) was purchased from Selleckchem (Cat# S4430). Actinomycin D was purchased from Sigma (Cat: A1410-5MG). Recombinant HnRNPK (Cat# ab132460) recombinant CBFB (Cat# ab98252) and recombinant eIF4B (Cat# ab114511) proteins were procured from Abcam. ARCA (Anti-Reverse Cap Analog, Cat# AM8045) was purchased from Invitrogen.

For immunoblotting, following antibodies were used: CBFB (Bethyl, Cat:A303-547A, 1:1000), RUNX1 (Cell signaling, Cat:4334 s, 1:1000), RUNX2 (cell signaling, Cat:8486, 1:1000), RUNX3 (Cell signaling, Cat: 9647 S, 1:1000), hnRNPK (Bethyl, Cat: A300-674A, Ab1, 1:1000; Cat: A300-675A, Ab2, 1:1000; Cat: A300-676A, Ab3, 1:1000), NOTCH3 (cell signaling, Cat:5276 s, 1:1000),), LC3A/B (cell signaling, Cat:12741 s, 1:1000), β actin (Sigma, Cat:A5316, 1:5000), H3 (Millipore, Cat:07-690:, 1:5000), GAPDH (Abcam, Cat:ab9484, 1:5000), HnRNPE2 (Cell signaling, Cat:83017, 1:1000), eIF4B (Cell signaling, Cat: 13088 S, 1:1000), SET1/COMPASS Antibody Sampler Kit (Cell signaling, Cat:25501 T, 1:1000), FRA1 (Cell signaling, Cat:5281 S, 1:1000), TCF1/TCF7(Cell signaling,Cat:2203 S, 1:1000), TGF-β (Cell signaling, Cat:3709 S, 1:1000), Drosha (Cell signaling, Cat:3364 S, 1:1000), GSK-3α (Cell signaling, Cat:4818 S,1:1000), SUFU (Cell signaling, Cat: 2522 S, 1:1000), PKM1/2 (Cell signaling, Cat:3106 S, 1:1000), NUMBL Antibody AbVantage™ Pack (Bethyl, Cat: A310-611A, 1:1000) and Translation Initiation Complex Antibody Sampler Kit (Cell signaling, Cat: 4763, 1:1000). For Chromatin immunoprecipitation (ChIP), we used RUNX1 (Abcam, Cat: Ab23980,10 μg). For Immunohistochemistry (IHC), we used antibodies for CBFB (Bethyl, Cat: A393-549A, 1:100), RUNX1 (Abcam, Cat: ab23980, 1:100), hnRNPK (Bethyl, Cat: A300-674A, 1:100), and NOTCH3 (Abcam, Cat: ab23426, 1:100). For Immunofluorescence (IFC), we used following antibodies: CBFB (Bethyl, Cat: A303-549A, 1:100), RUNX1(Abcam, Cat: Ab23980,1:100), hnRNPK (Bethyl, Cat: A300-674A, 1:100; Cat: A300-675A, 1:100; Cat: A300-676A, 1:100). For RNA immunoprecipitation (RIP) we used antibodies: CBFB (Bethyl, Cat: A303-548A, 1 μg), HnRNPK (Bethyl, Cat: A303-674A, 1 μg), HnRNPL (Bethyl, Cat: A311-423A, 1 μg), HnRNPE2 (Abcam, Cat: ab184962, 1 μg) and eIF4B (Bethyl, Cat: A301-766A, 1 μg).

**Cell culture**. MCF10A, MCF7, BT474, T47D, ZR751, MDA-MB-157, MDA-MB-231, MDA-MB-361, MDA-MB-436, MDA-MB-453, MDA-MB-468 and HEK-293T cells were purchased from ATCC (Manassas, VA) and are mycoplasma free. MCF12A cells were a kind gift from Stefan Ambs (NIH, Bethesda). MCF7E (early passage MCF7 cells) were a gift from Michael G. Brattain, and were authenticated by short terminal repeat (STR) analysis and shown to be mycoplasma free[31]. MCF7E cells were grown in Eagle's Minimum Essential Medium supplemented with 10% FBS. MCF10A and MCF12A were cultured in DMEM/F12 media supplemented with 5% horse serum, 10 μg/ml human recombinant insulin, 20 ng/ml human EGF, 500 ng/ml Hydrocortisone, 100 ng/ml Cholera toxin and antibiotics.

MCF7 were cultured in ATCC formulated Eagle's Minimum Essential Medium (EMEM) supplemented with 10% FBS and antibiotics. All other cell lines were cultured in DMEM + 10% FBS media supplemented with antibiotics. All cell lines were maintained at 37 °C in an incubator supplied with 5% CO$_2$.

**Lentivirus production, cloning, and site-directed mutagenesis.** For Lentivirus production, plasmids of interest with lentiviral backbone and packaging vectors pCMV deltaR8 (Addgene, Cat:12262) and pCMV VsVg (Addgene, Cat:8454) were transfected into LentiX-293T cells. After transfection, supernatant was collected at 48 h and 72 h time period. These two bathes of supernatants were combined and used to transduce cells of interest with polybrene (Fluka, Cat:52495, 6 μg/ml). Finally, transduced cells were selected using Blasticidin (Invitrogen, Cat: R210-01, 10 μg/ml) or Puromycin (Gemini Bio-products, Cat:400-128p, 2 μg/ml) depending on plasmid backbone. Oligos used in this study are in the Supplementary Data 2. We used pLenti6/V5 and pCW57.1 based vector systems for expressing a gene. Site-directed mutagenesis was performed using Agilent's QuikChange kit or Tag-Master® Site-Directed Mutagenesis kit (GM biosciences, Rockville, USA, Cat# 7001) as per manufacturer's instructions.

**Immunofluorescence staining and confocal microscopy.** $2 \times 10^5$ MCF10A cells were plated overnight in Lab-Tek II chamber slide (2 well, Nunc, Cat# 155380). Next day, cells were fixed with 4% formaldehyde in PBS for 15 min at room temperature (RT). Subsequently cells were washed 3 times with PBS and incubated in blocking buffer (5% normal serum + 0.3% Triton-X100 in 1X PBS) for 1 h at RT. Further, cells were incubated overnight in primary antibody for CBFB, RUNX1 or hnRNPK at 4 °C. Next day, cells were extensively washed with PBS followed by incubation in secondary antibody anti-rabbit Alexa 488 (1:500) at RT for 2 h. After incubation, cells were washed with PBS and incubated with DAPI (1:2000) for 1 h at RT for visualization of nuclei. After staining, cells were washed and mounted with VECTASHIELD® mounting media (Vector laboratories, Inc Burlingame, CA). Confocal microscopy of fluorescence stained cells was performed on Zeiss LSM 780 microscope at 63X oil immersion objective lens.

**Immunohistochemistry.** Slides with formalin fixed and paraffin embedded tissue sections were deparaffinized using Xylene and serially hydrated by incubating in decreasing percentage of alcohol (100 to 50%). Antigens were retrieved by boiling in 10 mM sodium citrate for 20 min. After cooling, endogenous peroxidases were deactivated by incubating in 3% H$_2$O$_2$ for 10 min. Subsequently, slides were washed with PBS + 0.1% Tween 20 and blocked with serum. After blocking, slides were incubated overnight with primary antibodies for CBFB, RUNX1, NOTCH3, or hNRNPK at 4 °C. Next day, after washing with PBS, slides were incubated in biotinylated goat anti-rabbit secondary antibody (VECTASTAIN ABC kit) for 30 min. Further, slides were incubated in biotin avidin solution for 30 min at RT. Finally, color was developed by incubating in DAB for 5–10 min.

**CRISPR knockout and single clone selection.** CRISPR targeting sequences (see Supplementary Data 2) for corresponding genes (CBFB, RUNX1, NOTCH3) were cloned into either PX330 vector (Addgene# 42230) or LentiCRISPRV2 vector (Addgene# 52961). For gene deletion, CRISPR constructs were co-transfected with eGFP in cells. eGFP positive cells were sorted by flow cytometry and plated at a low density (200 cells/10 cm plate) to obtain single cell colonies. Single cell colonies were genotyped by PCR and western blotting.

**FLAG pull down and mass spectroscopy analysis.** We established stable CBFB KO MCF10A cell line expressing empty vector (EV, control), CBFB_FLAG (C-terminal FLAG) or FLAG_CBFB (N-terminal FLAG) by transducing cells with lentivirus (plenti6-GW/V5, plenti6-CBFB_FLAG or plenti6-GW_FLAG_CBFB). We selected transduced cells with 10 μg/ml Blasticidin. Around 100 million stably transduced cells were used to perform FLAG pulldowns shown previously[14,32]. Briefly, cells were lysed in 10 ml of NET buffer (50 mM Tris, pH 7.5, 250 mM NaCl, 5 mM EDTA, 0.1% NP40 plus protease inhibitors, 10 μg/ml RNase A). Cell lysate was incubated with 200 μl of anti-FLAG-M2 affinity gel (Sigma, Cat# A2220) overnight at 4 °C. The next day, FLAG-M2 beads were washed extensively with NET buffer and eluted 4 times with 500 μg/ml of 3x FLAG peptide (Sigma, Cat#F4799). Eluted proteins were combined, precipitated with acetone, resolved using the NuPAGE 4–12% Bis-Tris protein gels (ThermoFisher, Cat: NP0336BOX) and stained using the Silver Quest Staining Kit (ThermoFisher, Cat# LC6070). We cut out enriched bands (C-terminal Flag-CBFB or N-terminal Flag-CBFB vs. empty vector) and subject them to mass spectrometry analysis. Therefore, this approach is not a comprehensive survey of all the interactors of CBFB. Instead, it preferentially detects CBFB interactors with a high stoichiometry ratio to CBFB. Mass spectrometry analysis was performed at NCI protein laboratory.

**Anchorage independent assay (soft agar colony assay).** For soft agar colony assay, we prepared a bedding of 0.5% agarose (Sea Kem® LE Agarose, Cat#50004) in culture medium in 6 well plates and allowed the agarose to solidify completely. Subsequently, 2,000 cells were mixed with warm 0.35% agarose in culture medium and layered on to top of the bedding. At day 15 and 30, number of cell colonies was

counted and imaged at ×20. The percentage of transformed cells was calculated as the number of colonies divided by the total number of plated cells. Colony size was calculated based on images of 20 randomly selected colonies (duplicates of wells, 10 images from colonies in each well). Total three biological repeats were performed for each experiment.

**RNA pulldown assay (RPA).** For RNA pulldown assay, *RUNX1* mRNA fragments were cloned into pBSKS vector and in vitro transcribed to label mRNA with biotin. For RPA, 20 ul of M-280 Dynabeads (Cat: 11205D, ThermoFisher) were incubated overnight with 600 ng of biotinylated mRNA. Next day, Dynabeads/mRNA complexes were washed and incubated with 400 μg of whole cell lysate at 4 °C for 1.5 h. Dynabeads were washed with RPA buffer (50 mM Tris, pH7.5, 250 mM KCl, 5 mM EDTA, 0.5 mM DTT, 1% NP40 plus protease inhibitors and RNase inhibitors), and interacting proteins were analyzed using western blotting.

**RNA immunoprecipitation (RIP).** For RNA immunoprecipitation, 1 μg of CBFB, HnRNPK, or HnRNPL antibody was incubated overnight with 500 μg of cell lysate at 4 °C. Next day, protein A agarose beads were added to lysate and incubated for additional 4 h at 4 °C. Finally, beads were washed with RPA buffer and total RNA was extracted using Trizol (Invitrogen) and analyzed by real-time PCR.

**In vitro translation assay.** Most in vitro translation assays were performed using the 1-Step Human coupled IVT kit (ThermoFisher Scientific, Cat# 88882) unless otherwise indicated in the figures. Briefly, DNA sequences that encode RNA were cloned into the pT7CFE1-CHis vector (for cap-independent translation) or the pBSKS vector (for cap-dependent translation). RNA was generated by in vitro transcription and was purified using RNeasy Mini Kit (Qiagen, Cat: 74104) as previously described[32]. For in vitro translation reaction, 31 ng of RNA was incubated with components in the IVT kit as per manufacturer's instructions. At the end of incubation, translated protein was visualized by Western blotting.

For cap-independent translation assays (Figs 4b, c, 6b lower panel), DNA was cloned into the pT7CFE1-CHis vector, which contains the Encephalomyocarditis virus (EMCV) internal ribosome entry site (IRES) element, bypassing the requirement for the m$^7$G cap. For cap-dependent translation assay (Fig. 6c upper panel), DNA was cloned into the pBSKS vector. The m$^7$G cap was added during the in vitro transcription step using T7 polymerase. Briefly, anti-reverse cap analog (ARCA, ThermoFisher, Cat: AM8045) was added into the reactions in the TranscriptAid T7 High Yield Transcription Kit (ThermoFisher, Cat: K0441) with an ARCA:GTP ratio of 4:1. RNAs generated by in vitro transcription were used in the in vitro translation kit, which uses HeLa cell lysate (ThermoFisher, Cat# 88882).

Recombinant CBFB (Abcam: Cat: ab98252), hnRNPK (Abcam, Cat: ab98252), and eIF4B (Abcam, Cat: ab114511) were used in the in vitro translation.

For in vitro translation assay described in Fig. 4d, e, HEK-293T cell lysate was prepared as previously described (Witherell, 2001). Briefly, CBFB WT, CBFB_KO, hnRNPK KD HEK-293T cells were grown in 15 × 15-cm dishes until confluency. To prepare whole cell extract, cells were incubated in a hypotonic buffer (10 mM HEPES-KOH, pH7.4, 10 mM KOAc2, 0.5 mM MgOAc2, 5 mM DTT and protease inhibitors (EDTA free) in a 1.5 times volume of cell pellet. Cells were lysed in a Dounce homogenizer with 40 firm strokes. Afterwards, homogenate was centrifuged at 12,000 × g for 10 min, and supernatants were flash frozen before stored in −80 °C. In vitro translation reaction was set up with 100 ng of RNA containing EMCV internal ribosome entry site (cap-independent), 50% cell extract (prepared from HEK-293T cells as mentioned above), 60 μM amino acids, 0.8 mM ATP, 0.1 mM GTP, 16 mM HEPES-KOH pH 7.4, 20 mM creatine phosphate, 40 μg/ml creatine phosphokinase, 50 μM Spermidine, 3 mM MgOAc2 and 160 mM KOAc2 for 3 h at 30 °C. Translated proteins were visualized by western blotting.

**ChIPseq, RNAseq, and data analysis.** ChIPseq and RNAseq were performed at the Next Generation Sequencing Facility, Centre for Cancer Research at NCI. For ChIPseq, 10 ng of IP DNA sample was subjected to deep sequencing. Peaks were identified using the MACS algorithm[33]. For RUNX1 ChIPseq, we first performed an endogenous ChIPseq and we did not see strong binding of RUNX1 on chromatin. This could be due to the low expression of RUNX1 and/or low affinity of the RUNX1 antibody. To test these possibilities, we used a stable MCF10A cell line in which RUNX1 was deleted and a doxycycline inducible RUNX1 vector was stably integrated. Using this RUNX1 inducible cell line, we performed ChIPseq of RUNX1 after treating the cells with 1 μg/ml doxycycline for 24 h and used the dataset for this study. For RNAseq 1 μg of total RNA was subjected to rRNA removal, size selection, cluster generation and high throughput sequencing on the Hiseq2500 platform.

**Subcellular fractionation.** To separate cytoplasmic and nuclear fractions, different cells were fractionated using the PARIS™ kit (ThermoFisher Scientific, Cat#AM1921) as per manufacturer's instructions.

**RNAseq analysis.** We applied the DESeq algorithm[34] to compare the RUNX1 knockout MCF10A cells vs. wild type cells, and the CBFB knockout cells vs. wild

type cells. We obtained two gene sets at FDR = 0.05 and fold-change threshold 2. There were 212 genes in the intersect of the two gene sets in which RUNX1 knockouts and CBFB knockouts had the same up and down expression change directions compared to the wild type samples.

**Application of the gene signature to human breast cancer prognosis**. From the 74 genes downregulated in both RNUX1 knockouts and CBFB knockouts (Thus, these genes are upregulated by CBFB and RUNX1.Hereinafter, they are called CBFB- and RUNX1-induced genes), 51 genes have gene expression for $n = 995$ patients with survival information in the TCGA breast cancer dataset. We found the top 10 out of the 51 genes induced by CBFB and RUNX1 gave the best prognosis result. These 10 genes include ROS1, ANO1, PNLIPRP3, DMBT1, MCF2, FHOD3, GDA, RNF165, B3GALNT1, MARC2. We then applied the 10-gene signature to another cohort, METABRIC, with about 2000 breast cancer patients. Those patients with high signature had better survival.

**Polysome fractionation**. Polyribosome profiling was performed with modifications from a previously described protocol[35]. Specifically, 2 million WT and CBFB_KO MCF10A cells were treated with 100 µg/ml of cycloheximide at 37 °C for 30 min. Afterwards, cells were washed with chilled PBS and harvested with trypsin. Subsequently, cells were lysed in polysome lysis buffer (20 mM Tris, pH7.2, 130 mM KCl, 15 mM MgCl₂, 2.5 mM DTT, 0.5% NP-40, 0.2 mg/ml Heparin, 0.5% deoxycholic acid, 100 µg/ml cycloheximide supplemented with protease inhibitors and RNase inhibitors) for 20 min at 4 °C. Cleared lysate was prepared by centrifuging samples at $8000 \times g$ for 10 min. Further, lysate was carefully layered upon 10–50% linear sucrose gradient and ultracentrifuged at $100,000 \times g$ at 4 °C for 2 h. After centrifuging, fractions were obtained using fractionator (Biocomp Instruments, Fredericton, Canada). RNA was isolated from individual fractions using TRIzol (Invitrogen) and subject to either real-time PCR or RNAseq.

**Statistics**. Number of repeats (n) were described in figure legends. Statistical significance was calculated using two-tailed $t$ test (Figs 2d, e, 3d, 5c–f, 7g tumor weight), chi-square test (Fig. 5g), and Fisher exact test (Fig. 7g tumor incidence).

**Animal studies**. Mice were maintained under the guidelines of Institutional Animal Care and Use Committee (IACUC)-approved protocols of National Cancer Institute (NCI) and National Heart, Lung and Blood Institute (NHLBI). Animal care was provided in accordance with the procedures outlined in the "Guide for the Care and Use of Laboratory Animals", National Research Council; 2011 National Academies Press; Washington DC.

For experiments using MCF10A cells and NSG mice, five million cells were resuspended in DMEM/F12 media plus 25 mM HEPES and mixed with 50 µl Matrigel before being transplanted subcutaneously into NSG mice. After 60–80 days, tumors, if any, were harvested, weighed and fixed in 10% neutral buffered formalin for 16 h and subsequently processed for Hematoxylin and Eosin staining by Histoserv, Inc (Germantown, MD, USA). H&E sections of xenografts were reviewed in an independent, blinded fashion. Tissues were evaluated based on histologic features (differentiation states and epithelial to mesenchymal differentiation for example), inflammation within tumors, extent of necrosis, invasion by tumor into surrounding non-neoplastic tissues, and evidence of metastasis within tissues.

For experiments using MCF7E cells and athymic nu/nu mice (Nude mice), mice were supplemented with 60-day 0.72 mg slow release 17β-estradiol pellets purchased from Innovative Research of America (IRA). Cells in PBS ($5 \times 10^5$) were injected into the #2 mammary fat pad of 6–8 weeks old female virgin nude mice. Tumor growth was monitored over time using electronic calipers. The greatest longitudinal diameter (length) and the greatest transverse diameter (width) were measured. Tumor volumes were estimated by the modified ellipsoidal formula: volume = 1 / 2(length × width × width)[36]. After 60 days, the mice were euthanized by carbon dioxide narcosis. Primary tumors were excised, bisected and snap frozen in liquid nitrogen for molecular analyses, or fixed in 10% neutral buffered formalin (NBF).

**Human breast cancer tissue microarray**. Human tissue microarrays (TMA) of breast cancer and normal tissue were purchased from US Biomax, Inc. (Cat# BC081120c, BRN801b). Breast cancer TMA slide (BC081120c) contains 110 cases/ 110 cores (100 breast tumors and 10 adjacent normal breast tissues) including pathology grade. Normal breast tissue TMA (BRN801b) has 80 cases/80 cores (10 breast tumors and 70 normal breast tissues).

**Ethical approval**. This study does not involve experiments that require ethical approval.

Original data are provided in a Source Data file

**Reporting Summary**. Further information on research design is available in the Nature Research Reporting Summary linked to this article.

## Data availability

Genomic data were submitted to the Gene Expression Omnibus (GEO) database with an accession number: GSE119131 (Polysome profiling), GSE120216 (RNAseq of CBFB WT, KO, RUNX1 WT and KO), GSE119800 (RIPseq of CBFB and hnRNPK). GSE129314 (RUNX1 ChIPseq). The source data underlying Figs. 1a, 1c-d, 1f, 1h-j, 2b-f, 2h-i, 3d, 4b-g, 5c, 5e-f, 6a-d, 7c, 7e-f, 7h-i and Supplementary Figs 1a-b, 1d-h, 3a-b, 3d-f, 3h, 4b-f, 5a, and 6a-l, 7a-i, 8b-e, 8g, 8i-k, 9d are provided as a Source Data file. All the other data supporting the findings of this study are available within the article and its Supplementary Information files and from the corresponding author upon reasonable request. A reporting summary for this article is available as a Supplementary Information file.

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

## Acknowledgements

This work was supported by the National Cancer Institute, USA, intramural grant, 1ZIABC011504-05 to Jing Huang. We thank Alan Hinnebusch for insightful suggestions on translation regulation, Bao Tran's Next Generation Sequencing Facility at the Center for Cancer Research (CCR) for RNA-seq, the Office of Science and Technology Resources (OSTR) at CCR, NCI and NCI director's innovation award for partial funding support.

## Author contributions

N. Malik, H. Yan, N. Moshkovich, M.P., V.S., Z.C., S.J. and C.L. performed the experiments and/or contributed key reagents; N. Malik, H. Yang, T.P., M.L., L.W. and J.H. analyzed the data; B.M., S.Y., D.L., L.W. and J.H. contribute to experiment design and data analysis; N. Malik and J.H. conceived the concept and wrote the manuscript; everyone read and commented on the manuscript.

## Additional information

**Competing interests:** The authors declare no competing interests.

