## [Peer Review File · Nature Communications]

Reviewers' comments:

Reviewer #1 (Remarks to the Author):

In the manuscript Malik and colleagues identify a role for CFBF in translation regulation. This is a very surprising discovery since CFBF has previously been widely implicated as a transcription factor involved in transcriptional regulation together with RUNX proteins. Using various models the authors show that CFBF has cytoplasmic localization and acts in a complex with hnRNPK and eIF4B to bind a large set of mRNAs amongst which the RUNX1 mRNA. In addition, the authors show that CFBF in complex with RUNX1 still has a role in transcriptional regulation, suggesting that both transcription and translation are affected upon CFBF mutation. Together, these results provide some very interesting novel suggestions on how core binding factor complexes operate and might be deregulated upon mutation of one of its components. In that respect it would be interesting if the authors can also speculate on RUNX mutations in breast cancer. Are these affecting the RUNX1-CFBF interaction and alter the cytoplasmic/nuclear ratio of CFBF? Also, based on the knockout analysis the authors suggest that the role of CFBF in the cytoplasm is as important as its role in the nucleus. However, in patients (figure 1f) it seems as if reduced CFBF (one allele mutated) is sufficient to shut down RUNX1 expression, suggesting the effects might be more dramatic for transcription. It would also be interesting to know what the interaction domain of CFBF with hnRNPK is. CFBF can be expressed as a fusion in hematological diseases, and the question is whether the same mechanism could be operational. Additional points include:

- + Can the authors list all CFBF interactors identified by mass spectrometry in a table?
- + Core binding factors interactomes have been extensively studied, did any of these previous studies identify an interaction of core binding factors with hnRNP or eIF factors?
- + Repair the mistakes throughout the text. Some figures are not discussed or wrongly labeled, for example supplementary fig 2a is not discussed; on page 10 there are wrong figure references (5c and 5d); page 11 figure 4c should be supplementary figure 4B; the supplementary figure 2 legend is not correct for the last panels; the y-axis of figure 5B: total instead of toal; remove '(continued)' on page 4 of the supplementary part.

Reviewer #2 (Remarks to the Author):

Major concerns:

The biological significance of CFBF in translation control needs further detail and substantiation. There is insufficient data demonstrating the role of the dysregulation of this axis in breast tumor progression. While the last figure regarding the nuclear function of CFBF is of note, the biological impact of this axis needs to be demonstrated.

The part of nuclear CFBF/RUNX1 regulation of NOTCH3 is too preliminary and needs further details and experimentation.

Minor Concerns:

Although the "empty vector condition is well used, there is no IgG controls on IP over the study for protein/protein IPs.

Figure 2j is useless if not more detailed.

Figure 3a and b: the cutoff setup to define the enrichment needs to be detailed in the dot plot by coloring the dots above the cutoff or by using other plotting system.

Figure 3 a and b: what might be the points with high FPKM and down regulated in FLAG-Protein conditions? Especially on FLAG-CBFB RIP

Figure 3b: Is the principal regression line of non-bound transcripts on equation $y=1*x$? It seems to be not properly axed but rather shifted to the input axis. What could be the reason?

The legend of Figure 3 needs more detail.

Figure 4c the authors conclude "and recombinant hnRNPK greatly potentiated the effect of recombinant CBFB. This is not apparent from the data presented.

In Figure 6, how was eIF4B identified as a binding partner of CBFB?

The M&M section need much more detail.

Reviewer #3 (Remarks to the Author):

In the manuscript entitled "The transcription factor CBFB suppresses breast cancer through orchestrating translation and transcription", the authors attempt to determine what role CBFB (and its subsequent loss) plays in regulating gene expression and what impact this has on breast cancer. Through these studies, the authors, make a concerted effort to understand how CBFB regulates its transcriptional partner, RUNX1, (along with several other genes), through its interaction with these respective mRNA transcripts.

Through these studies, Malik et al, delineates a unique and novel role for CBFB in regulating translation (in addition to its canonical novel role in transcription) with an RNA binding protein HNRNPK.

This manuscripts offers several novel findings and provides evidence for an undescribed role for CBFB in translation.

Major Issues:

-In Figure 1B, the tumors isolated from mice transplanted with CBFB-KO cell lines should have a corresponding CBFB and most importantly RUNX1 IHC images. The RUNX1 data are critical for evaluating the more in vivo nature of this relationship. If these are injections into a mammary fat pads, adjacent "normal" tissue would serve as an excellent control.

-In Figure 1H, the authors show that while RUNX1 knock down does not affect total CBFB levels, it does decrease CBFB nuclear localization. It would be helpful to have an immunofluorescence image corroborating the western blot.

-In Figure 1i, the authors demonstrate that the N-terminal Flag-tagging of CBFB abrogates its normal function (ie- its interaction with RUNX1). This would indicate this protein is not fully functional or is not properly folded or has sterically hindrances. However, this construct is then used to identify novel interacting proteins in Mass-Spec and further used in Co-IP studies to assess its interactions with HNRNPK and eIF4B (Figures 2b, Figure 6a). For the Mass-Spec, the Material and Methods Sections states that the experiments were performed with both the N- and C-terminal CBFB. Thus, the silver stain showing the interactions with HNRNPK and the C-terminal CBFB should be included. Also, as a minimum, the Co-IP experiments should be repeated with a C-terminal tagged form of CBFB to validate that these interaction occur with HNRNPK and eIF4b.

-Additionally, the Mass-Spec experiments appear to have superphysiologic levels of CBFB. In fact, the EV lane appears devoid of a CBFB band. This raises the question of whether CBFB interacts with hnRNP K in a native setting. Therefore, the authors should include a Co-IP data from

endogenous lysates demonstrating an interaction between 1) CBFB and HNRNPK and 2) CBFB and eIF4B.

- Given the manuscript's focus on the role of CBFB in regulating RUNX1 translation via hnRNP K in the cytoplasm. It would be useful for the authors to show levels of interaction between endogenous CBFB and hnRNP K in the cytoplasm and nucleus by Co-IP of fractionated lysates.

- Since the N-terminal Flag-CBFB is unable to interact with RUNX1, does it actually localize to the nucleus or is it sequestered to the cytoplasm. Flag immunofluorescence should be included to determine its ability to localize in both the cytoplasm and nucleus.

- Throughout Figure 2, the authors present data showing an elaborate feedback loop between differing proteins but have not addressed whether altering CBFB expression impacts HNRNPK expression. Since the conclusions of this study rely on HNRNPK acting as an intermediate, this is an important consideration to evaluate. Likewise, does altering hnRNP K expression alter CBFB expression? Western blots could be used to evaluate any changes (or lack of changes) in the CBFB-KO and HNRNPK-KD cells.

- In many published reports, the human protein atlas, and product data sheets, HNRNPK is shown to be a primarily nuclear protein (by IF and IHC) with some increased cytoplasmic staining in some cancers and when serum starved or following activation of the MAPK pathway (PMID: 11231586). However, in the fractionation studies presented in Supp 3A, the majority or the near majority of HNRNPK is cytoplasmic. This is even the case of the widely used MCF7 cells. These events appear out of the norm. Thus, to better understand the results of these study and the interactions between CBFB and HNRNPK, immunofluorescence with additional HNRNPK antibodies are needed on several cell lines, including non-transformed/normal, luminal, and TNBC cell lines.

- Also, but perhaps more important for supporting the main conclusions of the manuscript, is the need for HNRNPK levels and subcellular localization to be examined by IHC in normal breast tissues (TMA) so that the ratio of cytoplasmic to nuclear staining can be evaluated. This is critical for the authors so that they can more fully conclude this is in fact a general mechanism.

- I assume this is a "copy/paste" oversight, but the PMK1/2 panel in 4F appears to be a duplicated image from 4G. This is evident in particular because there are three lanes in the CBFB knockdown images, but this particular blot only has two lanes.

- In Supp Figure 3E, the authors use HNRNPL as a control in their RIP experiments. It is not clear why the authors picked this RNA-binding protein as a control. HNRNPK has KH domains while hnRNP L has RRM domains for binding to RNA. Even though the RNA binding motifs for these two RNA binding proteins include Cs, but are still vastly different. The use of a more appropriate RNA-binding control should be used in this setting (HNRNPE2 as an example).

- In the polysome trace in Figure 5A, there are significant global changes in the 40S, 60S and polysome fraction in the CBFB KO cells. It appears that the entire absorbance curve is shifted upwards for CBFB KO samples. Is this because more RNA was present in the CBFB KO cells? Since the Material and Methods indicate that an equal number of cells were used, is this reflective of CBFB's role in transcription and RNA processing, wherein a knockout of CBFB results in higher RNA levels. Thus, a global role for CBFB would alter the interpretation of the polysome data. Using your CBFB-KO RNA-Seq data, can the authors determine if there are global changes in total RNA expression compared to WT? Regardless of the outcome, this would assist in determining whether CBFB plays an increased role in global translation.

- The authors indicate that the distribution of GAPDH mRNA does not change upon CFBF deletion. However, there appears to be significant differences between fractions in the light polysomes and heavy polysomes (Fractions 9-12 and 13-15). Similar to the point above, does this indicate that CFBF may also be playing a critical role in global translation? If these fractions are not different, please briefly indicate in the text why this result is insignificant as it could confuse a general reader who does not perform this type of assay. If these data are different, could the authors briefly elaborate on a potential additional novel role for CFBF in global translation?

- A western blot depicting the different polysome fractions using ribosomal control proteins is needed to evaluate the separation of the different ribosomal components.

- In Figure 6B, the authors do not have a control sample containing a combination of HNRNPK and eIF4B to assess their effect on RUNX1 protein levels. This is a critical control and would also indicate whether HNRNPK and eIF4B can function independent of CFBF to stimulate RUNX1 translation, indicating that CFBF may not be the only upstream signal sufficient for stimulating RUNX1 translation. This control should be included.

- Given the described role of HNRNPK as a transcription factor, and data in this manuscript demonstrating an interaction between HNRNPK and CFBF, did the authors examine a role for HNRNPK/CFBF-RUNX1 in Notch signaling? Do these three proteins Co-IP in a complex? Do they CHIP to Notch3 or any other global target?

- The authors demonstrate in Supp 5A and 7B that down modulation of CFBF/RUNX1 targets portent for poor outcomes. While important, these analyses do not directly address CFBF loss in breast cancer. Given the large TMA in Supp Figure 6, the authors should evaluate clinical correlates (ie- survival analyses) on the patient samples comparing no CFBF to + CFBF expression or not-detected/low to Mid/High expression. If a significant number of patients are living with disease or in remission, other clinical responses (pathologic, metastatic, GRADE, etc) should then be performed. Without such analyses, these studies have a much weaker clinical relevance.

Further, given the relationship between CFBF and RUNX1 and Notch3 expression, IHC of RUNX1 and Notch3 levels from serial sections of the TMA must be included. This will provide clinical evidence outside of a cell based system for this novel discovery. Additionally, these results can then be readily stratified with the aforementioned survival analyses to confirm this is a clinical relevant pathway.

Minor Issue:

-In the luminal breast cancer cell lines, RUNX1 expression is missing yet CFBF is present. Are there tumor type differences that account for this? The authors should made note of this (at least as a discussion point in Discussion Section).

-Were any other poly(C)-rich RBP (such as HNRNPE2, which binds to ssDNA/RNA sequence like HNRNPK evaluated for their interactions with the CFBF protein and RUNX1 RNA. If not, it may be useful to include a statement in the discussion that other RNA-binding proteins may also influence that these interactions as your data suggest CFBF may be the "rate-limiting factor".

- In Figure Supp 2D, the data presented show a statistical differences in expression in RUNX1, which runs counters the conclusion of the paper. However, these data do not appear different. Are they really significant?
- The Figure legends in Supp 2 are not accurate.
- Text mislabels figures 3c and 3d incorrectly—as 5c and 5d.
- Text should mention why the transcripts they validated further were chosen.
- What are the relative levels of CBFB and HNRNPK in the HeLa lysate that were used in the in vitro transcription assay? Are they comparable to 293T cells? Is a significant portion of HNRNPK localized in the cytoplasm in this cell line also? Immunofluorescence could be used.
- On Page 7, the authors state that CBFB does not regulates RUNX1 mRNA degradation. However, while RNA levels are not altered in their experiments, this manuscript does not present data showing RNA degradation or RNA half-life studies. This is particularly relevant, since the authors show that CBFB engages with an RNA binding protein at the 3'UTR of the RUNX1, and the 3'UTR of many mRNA often dictate stability. Thus to be accurate, and without any type of actinomycin studies it may be difficult to definitively state that degradation it not impacted.
- On page 10, the authors refer to segment T14 of the RUNX1 3'UTR as having enhancer activity. It enhances translation but is not an enhancer per se. Although minor, it would be useful to amend this statement since RUNX1 levels are controlled by enhancer elements and the original statement may cause confusion.

We are very grateful to our reviewers for their helpful comments and suggestions. Our point-by-point responses are as follows.

Reviewer #1

“Together, these results provide some very interesting novel suggestions on how core binding factor complexes operate and might be deregulated upon mutation of one of its components. In that respect it would be interesting if the authors can also speculate on RUNX mutations in breast cancer. Are these affecting the RUNX1-CBFB interaction and alter the cytoplasmic/nuclear ratio of CBFB?”

Response: We thank the reviewer for encouraging comments. To evaluate the roles of RUNX1 mutations, we generated three recurrent breast tumor-derived RUNX1 mutants (D96G*, L134R, and G143V). We first tested whether these mutants interact with CBFB and found D96G* and found that L134R did not while G143V did (Figure 1a for reviewers, attached at the end of this file). After transducing RUNX1 KO MCF10A cells with lentiviruses expressing WT or mutants, we found that only WT and G143V increased the nuclear localization of CBFB while D96G* and L134R did not. These results are very interesting. We chose to include these data at the end of this file because our study is focused on the novel function of CBFB.

“It would also be interesting to know what the interaction domain of CBFB with hnRNPK is.”

Response: To address this, we generated several CBFB truncations and tested their abilities to interact with hnRNPK. CBFB contains a domain called CBFB domain (Supplementary Figure 4e). We observed that CBFB truncations with shortened CBFB domain (T4, T5, T6) were not detectable probably due to instability (Supplementary Figure 4e). This result is consistent with our observation with CBFB mutations (Figure 1a), all of which are within the CBFB domain. CBFB T2 and T3 were detected at similar levels while T3 had reduced binding to hnRNPK, suggesting that CBFB binds to hnRNPK with the residues from 1 to 130.

We also mapped the region of hnRNPK that interacts with CBFB (Supplementary Figure 4f). hnRNPK has three KH domains. We found that the interacting domain of hnRNPK with CBFB fell within the first two KH domains (residues 1-220).

“CBFB can be expressed as a fusion in hematological diseases, and the question is whether the same mechanism could be operational.”

Response: This is an interesting question. As this study is focused on breast cancer, we did not study whether the same mechanism exists in hematopoietic cancers.

“Can the authors list all CBFB interactors identified by mass spectrometry in a table?”

Response: We have added a supplementary table (Supplementary Table 2) to list all the proteins identified by mass spectrometry. We note that these are the proteins identified by mass spectrometry using the enriched bands from silver staining (showing visual difference between the empty vector and Flag-CBFB sample). Thus, most of them need to be further validated by IP followed by immunoblotting.

“Core binding factors interactomes have been extensively studied, did any of these previous studies identify an interaction of core binding factors with hnRNP or eIF factors?”

Response: We interrogated the NCBI database and did not find hnRNP and eIF factors in the list of CBFY interactors. Therefore, the interaction between CBFY and hnRNPK and eIF4B is novel.

“Repair the mistakes throughout the text. Some figures are not discussed or wrongly labeled, for example supplementary fig 2a is not discussed; on page 10 there are wrong figure references (5c and 5d); page 11 figure 4c should be supplementary figure 4B; the supplementary figure 2 legend is not correct for the last panels; the y-axis of figure 5B: total instead of toal; remove ‘(continued)’ on page 4 of the supplementary part.”

Response: We thank the reviewer for pointing out these oversights. We have now corrected these errors and others to our best efforts.

Reviewer #2

“The biological significance of CFBF in translation control needs further detail and substantiation. There is insufficient data demonstrating the role of the dysregulation of this axis in breast tumor progression. While the last figure regarding the nuclear function of CFBF is of note, the biological impact of this axis needs to be demonstrated. The part of nuclear CFBF/RUNX1 regulation of NOTCH3 is too preliminary and needs further details and experimentation.”

Response: To further study the regulation of NOTCH3 by CFBF/RUNX1, we performed ChIPseq of RUNX1 to interrogate whether the regulation of CFBF/RUNX1 on NOTCH3 is direct or indirect. Our ChIPseq detected the binding of RUNX1 on the NOTCH3 locus, suggesting that the regulation of NOTCH3 by CFBF/RUNX1 is direct. Given that this regulation is conserved across multiple breast cell lines, we believe this regulation has important biological relevance in breast cancer. In addition, we performed CFBF, RUNX1, and NOTCH3 immunohistochemistry using the same breast cancer tissue microarray. We observed a reverse correlation of CFBF/RUNX1 IHC signal with NOTCH3 signal (Supplementary Figure 9e), suggesting that the regulation of NOTCH3 by CFBF/RUNX1 is conserved in human breast tumors.

Minor Concerns

“Although the “empty vector condition is well used, there is no IgG controls on IP over the study for protein/protein IPs.”

Response: We have now added several experiments using IgG in endogenous co-IP (Figure 6b and Supplementary Figure 4d, 6f).

“Figure 2j is useless if not more detailed.”

Response: We removed Figure 2j as the reviewer suggested.

“Figure 3a and b: the cutoff setup to define the enrichment needs to be detailed in the dot plot by coloring the dots above the cutoff or by using other plotting system.”

Response: We re-made the figures and colored the dots in red above the cutoff.

“Figure 3 a and b: what might be the points with high FPKM and down regulated in FLAGProtein conditions? Especially on FLAG-CFBF RIP”

Response: We assume that the reviewer was asking about what those genes that are beneath the enrichment cutoff line (in the previous submission and is removed in this version). If our understanding is correct, these genes are not enriched (see our answer to the next question).

*“Figure 3b: Is the principal regression line of non-bound transcripts on equation $y=1*x$? It seems to be not properly axed but rather shifted to the input axis. What could be the reason? The legend of Figure 3 needs more detail.”*

Response: In our previous submission: the red lines were enrichment cutoff lines but not regression lines. To increase clarity, we took reviewer’s suggestion and colored the enriched genes and removed the enrichment lines. We also added clarification in the legend of Figure 3.

“Figure 4c the authors conclude “and recombinant hnRNPk greatly potentiated the effect of recombinant CFBF. This is not apparent from the data presented.”

Response: In our previous version, we did not refer to the correct figure. The conclusion was in fact based on Supplementary Figure 7b (Lane 5 and 6). We have referenced the correct figure in this submission.

*“In Figure 6, how was eIF4B identified as a binding partner of CFBF?
The M&M section need much more detail.”*

Response: As suggested by the reviewer, we added more clarity into the M&M section. We performed FLAG IP as shown in Figure 2a. In addition to the enriched bands corresponding to hnRNPk, we also identified another enriched band and sent it for mass spectrometry analysis. All the identified proteins were described in the newly added Supplementary table 2. eIF4B was one of the proteins identified by mass spectrometry. We surveyed a panel of initiation factor included in the sampler kit from Cell Signaling Technology (Cat: 4763) and found that eIF4B is the strongest binder among these tested factors (Figure 6a).

Reviewer #3

“In the manuscript entitled “The transcription factor CFBF suppresses breast cancer through orchestrating translation and transcription”, the authors attempt to determine what role CFBF (and its subsequent loss) plays in regulating gene expression and what impact this has on breast cancer. Through these studies, the authors, make a concerted effort to understand how CFBF regulates its transcriptional partner, RUNX1, (along with several other genes), through its interaction with these respective mRNA transcripts.

Through these studies, Malik et al, delineates a unique and novel role for CFBF in regulating translation (in addition to its canonical novel role in transcription) with an RNA binding protein HNRNPK.

This manuscripts offers several novel findings and provides evidence for an undescribed role for CFBF in translation.”

Response: We are grateful for reviewer’s comments.

“In Figure 1B, the tumors isolated from mice transplanted with CFBF-KO cell lines should have a corresponding CFBF and most importantly RUNX1 IHC images. The RUNX1 data are critical for evaluating the more in vivo nature of this relationship. If these are injections into a mammary fat pads, adjacent “normal” tissue would serve as an excellent control.”

Response: We added CFBF and RUNX1 IHC images for CFBF_KO cell line-derived tumors (Supplementary Figure 2a). Note that we did not detect RUNX1 IHC signal because CFBF loss led to decreased levels of RUNX1 protein (Figure 1c). Our xenograft tumors were subcutaneously injected. Therefore, we could not use the adjacent normal tissue as a control. To overcome this, we included RUNX1 IHC staining using a tumor derived from KRAS driven MCF10A cells. In this positive control, we readily detected nuclear RUNX1 signal. In summary, CFBF controls RUNX1 protein levels *in vivo*.

“-In Figure 1H, the authors show that while RUNX1 knock down does not affect total CFBF levels, it does decrease CFBF nuclear localization. It would be helpful to have an immunofluorescence image corroborating the western blot.”

Response: We performed the immunofluorescence experiment as the reviewer suggested (Figure 2 for reviewers, at the end of this file). It appeared to us that CFBF levels decreased in RUNX1 KO cells. However, because the fraction of nuclear CFBF is much less compared to that of cytoplasmic CFBF (Figure 1g-h), we are not sure whether the immunofluorescence is sensitive enough for us to draw a confident conclusion. Therefore, we attached the data at the end of this file for the reviewer to evaluate.

“-In Figure 1i, the authors demonstrate that the N-terminal Flag-tagging of CFBF abrogates its normal function (ie- its interaction with RUNX1). This would indicate this protein is not fully functional or is not properly folded or has sterically hindrances. However, this construct is then used to identify novel interacting proteins in Mass-Spec and further used in Co-IP studies to assess its interactions with HNRNPK and eIF4B (Figures 2b, Figure 6a). For the Mass-Spec, the Material and Methods Sections states that the experiments were performed with both the N- and C-terminal CFBF. Thus, the silver stain showing the interactions with HNRNPK

and the C-terminal CFBF should be included. Also, as a minimum, the Co-IP experiments should be repeated with a C-terminal tagged form of CFBF to validate that these interaction occur with HNRNPK and eIF4b.”

Response: In this submission, we included the silver staining of FLAG IP using CFBF knockout MCF10A cells expressing C-terminal CFBF (Supplementary Figure 4a), mass spectrometry data (Supplementary Table 2), and immunoblotting validation of interaction between C-terminal CFBF and hnRNP K (Supplementary Figure 4b). In addition, we performed experiments of endogenous co-IP (Figure 6b and Supplementary Figure 4d). All these results support that CFBF interacts with hnRNP K and eIF4B.

“-Additionally, the Mass-Spec experiments appear to have superphysiologic levels of CFBF. In fact, the EV lane appears devoid of a CFBF band. This raises the question of whether CFBF interacts with hnRNP K in a native setting. Therefore, the authors should include a Co-IP data from endogenous lysates demonstrating an interaction between 1) CFBF and HNRNPK and 2) CFBF and eIF4B.”

Response: Because we used FLAG IP for Figure 2a, empty vector lane should not have CFBF. We added clarification in figure legends and Methods. In addition, we included co-IP using under endogenous condition (Figure 6b and Supplementary Figure 4d). Results show that CFBF interacts with hnRNP K and eIF4B in the endogenous setting.

“- Given the manuscript’s focus on the role of CFBF in regulating RUNX1 translation via hnRNP K in the cytoplasm. It would be useful for the authors to show levels of interaction between endogenous CFBF and hnRNP K in the cytoplasm and nucleus by Co-IP of fractionated lysates.”

Response: We included co-IP between endogenous CFBF and hnRNP K in the cytoplasm and nucleus (Supplementary Figure 4d). The result turned out to be very interesting. We only detected interaction of CFBF/hnRNP K in the cytoplasm but not in the nucleus.

“- Since the N-terminal Flag-CFBF is unable interact with RUNX1, does it actually localize to the nucleus or is it sequestered to the cytoplasm. Flag immunofluorescence should be included to determine its ability to localize in both the cytoplasm and nucleus.”

Response: We included immunofluorescence and fractionation data showing the localization of N-terminally and C-terminally FLAG tagged CFBF (Supplementary Figure 3c-d). N-terminal Flag-CFBF, like C-terminal Flag-CFBF, is mainly localized in the cytoplasm (Supplementary Figure 3c-d). We detected nuclear localization of FLAG-CFBF using immunoblotting (Supplementary figure 3d). Note that even in RUNX1 KO MCF10A cells, a portion of CFBF is still localized in the nucleus (Figure 1h), suggesting other mechanism(s) exists to shuttle CFBF into the nucleus.

“-Throughout Figure 2, the authors present data showing an elaborate feedback loop between differing proteins but have not addressed whether altering CFBF expression impacts HNRNPK expression. Since the conclusions of this study rely on HNRNPK acting as an intermediate, this is an important consideration to evaluate. Likewise, does altering hnRNP K expression alter CFBF expression? Western blots could be used to evaluate any changes (or lack of changes) in the CFBF-KO and HNRNPK-KD cells.”

Response: We performed immunoblotting of hnRNPk in WT and CBFb KO MCF10A cells (Supplementary Figure 4c). CBFb status did not alter the levels of hnRNPk. In Figure 2b (Input), overexpression of CBFb in CBFb_KO cells did not alter hnRNPk levels. Likewise, hnRNPk knockdown did not change the levels of CBFb (Figure 2c).

“- In many published reports, the human protein atlas, and product data sheets, HNRNPk is shown to be a primarily nuclear protein (by IF and IHC) with some increased cytoplasmic staining in some cancers and when serum starved or following activation of the MAPK pathway (PMID: 11231586). However, in the fractionation studies presented in Supp 3A, the majority or the near majority of HNRNPk is cytoplasmic. This is even the case of the widely used MCF7 cells. These events appear out of the norm. Thus, to better understand the results of these study and the interactions between CBFb and HNRNPk, immuno fluorescence with additional HNRNPk antibodies are needed on several cell lines, including nontransformed/ normal, luminal, and TNBC cell lines.”

Response: We note that we did not intend to claim that majority of hnRNPk is localized in the cytoplasm. In fact, our data showed that majority of hnRNPk is localized in the nucleus (Supplementary Figure 5a). We agree with the reviewers that the relative localization of hnRNPk in cytoplasm appears to be at the higher end of results reported in the literature.

To investigate systematically this, we performed fractionation and immunoblotting with three different hnRNPk antibodies (Bethyl Laboratories, Cat: A300-674A, A300-675A, A300-676A). We detected cytoplasmic localization of hnRNPk using the antibodies (Supplementary Figure 5a). Given that these three antibodies were generated using different epitopes within hnRNPk, the possibility of non-specific recognition is extremely low. Interestingly, in PMID 11231586, the authors also performed fractionation (their Figure 4a) and clearly showed that a substantial fraction of hnRNPk was in the cytoplasm.

To examine whether different methods could generate different results, we performed IF as the reviewer suggested using these three antibodies in several breast cell lines and detected hnRNPk both in the cytoplasm and nucleus in MCF10A, MCF7, and MDA-MB-468 cells (Supplementary Figure 5b). Interestingly, the portion of hnRNPk in the cytoplasm appeared to be less compared to results from cell fractionation. The different results generated from fractionation and IF methods were also reported in an independent study of hnRNPk (PMID 19170760), in which hnRNPk was found predominantly localized in the nucleus using IF while it was localized in both cytoplasm and nucleus using fractionation (their Figure 7). Therefore, it appears to be consistent that the cytoplasmic fraction of hnRNPk is under-detected in IF compared to the fractionation method. Why is that? It is conceivable that hnRNPk forms complex with binding partners in the cytoplasm. The fixation in IF crosslinks hnRNPk to its binding partners and masks (buries) its epitopes. In fractionation followed by immunoblotting, all proteins are denatured, and epitope is exposed. Although we cannot rule out other possibilities, this possibility is supported by emerging evidence showing that existence of membrane-less, granule-like structure in the cytoplasm that contains hnRNPk and protein translation machinery (PMID: 22579282). Further, we performed hnRNPk IHC using normal breast tissue and detected hnRNPk in both cytoplasm and nucleus (Supplementary Figure 5c). Note that a boiling step (reverse crosslinking) was included in our IHC protocol.

Regardless, since hnRNPk is abundantly expressed, a small fraction of hnRNPk in the cytoplasm may be enough to interact with CBFb, at least in breast cells. In summary, different

methods could lead to different interpretations of the ratio of hnRNPK in the cytoplasm versus nucleus. We hope these results and this discussion help other researchers studying hnRNPK.

“-Also, but perhaps more important for supporting the main conclusions of the manuscript, is the need for HNRNPK levels and subcellular localization to be examined by IHC in normal breast tissues (TMA) so that the ratio of cytoplasmic nuclear staining can be evaluated. This is critical for the authors so that they can more fully conclude this is in fact a general mechanism.”

Response: To address this, we performed IHC in normal breast tissues (TMA) (Supplementary Figure 5c). The result showed that hnRNPK was localized in both cytoplasm and nucleus.

“- I assume this is a “copy/paste” oversight, but the PMK1/2 panel in 4F appears to be a duplicated image from 4G. This is evident in particular because there are three lanes in the CFBF knockdown images, but this particular blot only has two lanes.”

Response: We thank the reviewer for pointing out this. We have corrected the figure.

“- In Supp Figure 3E, the authors use HNRNPL as a control in their RIP experiments. It is not clear why the authors picked this RNA-binding protein as a control. HNRNPK has KH domains while hnRNP L has RRM domains for binding to RNA. Even though the RNA binding motifs for these two RNA binding proteins include Cs, but are still vastly different. The use of a more appropriate RNA-binding control should be used in this setting (HNRNPE2 as an example).”

Response: Our laboratory had worked on hnRNPL before and we had the reagents for hnRNPL. We agree with the reviewer that another hnRNP needs to be used. We performed RIP using hnRNPE2 and observed interaction between hnRNPE2 and *RUNX1* mRNA (Supplementary Figure 6e). This result is probably not surprising given that hnRNPE2 is a polyC binding protein and *RUNX1* mRNA contains polyC tracts (Figure 2g). Two lines of evidence suggests that CFBF does not interact with *RUNX1* mRNA through hnRNPE2. First, the binding of hnRNPE2 to *RUNX1* mRNA was comparable between WT and CFBF KO MCF10A cells (Supplementary Figure 6e). Second, we could not detect interaction between hnRNPE2 and CFBF (Supplementary Figure 6f).

“- In the polysome trace in Figure 5A, there are significant global changes in the 40S, 60S and polysome fraction in the CFBF KO cells. It appears that the entire absorbance curve is shifted upwards for CFBF KO samples. Is this because more RNA was present in the CFBF KO cells? Since the Material and Methods indicate that an equal number of cells were used, is this reflective of CFBF’s role in transcription and RNA processing, wherein a knockout of CFBF results in higher RNA levels. Thus, a global role for CFBF would alter the interpretation of the polysome data. Using your CFBF-KO RNA-Seq data, can the authors determine if there are global changes in total RNA expression compared to WT? Regardless of the outcome, this would assist in determining whether CFBF plays an increased role in global translation.”

Response: We note that Figure 5a measures the protein absorbance instead of RNA. The appearing shift of CFBF KO cells was most likely due to machine baseline fluctuation, which is normal. Because RNAseq compares the relative abundance of each transcript, it is powerless to detect the global change of translation.

“- The authors indicate that the distribution of GAPDH mRNA does not change upon CFBF deletion. However, there appears to be significant differences between fractions in the light polysomes and heavy polysomes (Fractions 9-12 and 13-15). Similar to the point above, does this indicate that CFBF may also be playing a critical role in global translation? If these fractions are not different, please briefly indicate in the text why this result is insignificant as it could confuse a general reader who does not perform this type of assay. If these data are different, could the authors briefly elaborate on a potential additional novel role for CFBF in global translation?”

Response: We agree with the reviewer that for individual fractions in the polysome, *GAPDH* mRNA levels are different between WT and CFBF KO. However, *GAPDH* protein levels did not change between WT and CFBF KO cells, suggesting that the difference had no impact on the steady-state levels of *GAPDH* protein. In fact, all the polysomal fractions (actively translated mRNAs) are pooled and compared to the monosomal fraction (initiated but not actively translated mRNAs associated with ribosomes). We did not see the differences of ratios of poly- versus mono-ribosomal fractions between WT and CFBF KO cells for *GAPDH* mRNA. To clarify this, we modified the text.

“- A western blot depicting the different polysome fractions using ribosomal control proteins is needed to evaluate the separation of the different ribosomal components.”

Response: We added immunoblotting results of different polysome fractions using two ribosomal control proteins (RPS3 and RPL26) (Supplementary Figure 7i).

“- In Figure 6B, the authors do not have a control sample containing a combination of HNRNPK and eIF4B to assess their effect on RUNX1 protein levels. This is a critical control and would also indicate whether HNRNPK and eIF4B can function independent of CFBF to stimulate RUNX1 translation, indicating that CFBF may not be the only upstream signal sufficient for stimulating RUNX1 translation. This control should be included.”

Response: The experiment was repeated and included a condition of hnRNPk and eIF4B in the absence of CFBF. The combination of hnRNPk and eIF4B did not enhance the translation of *RUNX1* mRNA (Figure 6c, compare lane 2 and 5).

“- Given the described role of HNRNPK as a transcription factor, and data in this manuscript demonstrating an interaction between HNRNPK and CFBF, did the authors examine a role for HNRNPK/CFBF-RUNX1 in Notch signaling? Do these three proteins Co-IP in a complex? Do they ChIP to Notch3 or any other global target?”

Response: This is an interesting concept. We performed co-IP experiments and found that CFBF only interacts with hnRNPk in the cytoplasm but not in the nucleus (Supplementary Figure 4d). We speculate that in the nucleus CFBF preferentially binds to *RUNX1*. We also performed ChIPseq of *RUNX1* and detected binding of *RUNX1* on the *NOTCH3* locus (Figure 7d).

“- The authors demonstrate in Supp 5A and 7B that down modulation of CFBF/RUNX1 targets portent for poor outcomes. While important, these analyses do not directly address CFBF loss in breast cancer. Given the large TMA in Supp Figure 6, the authors should evaluate clinical correlates (ie- survival analyses) on the patient samples comparing no CFBF to + CFBF

expression or not-detected/low to Mid/High expression. If a significant number of patients are living with disease or in remission, other clinical responses (pathologic, metastatic, GRADE, etc) should then be performed. Without such analyses, these studies have a much weaker clinical relevance.”

Response: Our TMAs do not have clinical data such as survival, remission, and metastatic. They have tumor grade. We attempted to correlate the CFBF IHC signal to tumor grade and did not observe a significant correlation.

“Further, given the relationship between CFBF and RUNX1 and Notch3 expression, IHC of RUNX1 and Notch3 levels from serial sections of the TMA must be included. This will provide clinical evidence outside of a cell based system for this novel discovery. Additionally, these results can then be readily stratified with the aforementioned survival analyses to confirm this is a clinical relevant pathway.”

Response: As the reviewer suggested, we added RUNX1 and NOTCH3 IHC using serial sections of TMA (Supplementary Figure 9). Interestingly, we detected positive correlation between CFBF and RUNX1 IHC signals and negative correlation of these two proteins with NOTCH3. These results further support our molecular data showing CFBF and RUNX1 represses NOTCH3. As discuss above, our TMAs have no survival data.

Minor Issue:

“-In the luminal breast cancer cell lines, RUNX1 expression is missing yet CFBF is present. Are there tumor type differences that account for this? The authors should made note of this (at least as a discussion point in Discussion Section).”

Response: Indeed, RUNX1 levels are almost undetectable and CFBF levels are lower in luminal breast cancer cell lines. However, we felt that more detailed studies need to be done to draw a concrete conclusion. As the reviewer suggested, we discussed this point in the Discussion section.

“-Were any other poly(C)-rich RBP (such as HNRNPE2, which binds to ssDNA/RNA sequence like HNRNPK evaluated for their interactions with the CFBF protein and RUNX1 RNA. If not, it may be useful to include a statement in the discussion that other RNA-binding proteins may also influence that these interactions as your data suggest CFBF may be the “rate-limiting factor”.”

Response: As the reviewer suggested, we added experiments involving hnRNPE2. Using RIP, we detected interaction of hnRNPE2 and *RUNX1* mRNA (Supplementary Figure 6e). However, binding of hnRNPE2 to *RUNX1* mRNA in WT and CFBF KO cells was not different (Supplementary Figure 6e). In addition, hnRNPE2 did not interact with CFBF (Supplementary Figure 6f).

“- In Figure Supp 2D, the data presented show a statistical differences in expression in RUNX1, which runs counters the conclusion of the paper. However, these data do not appear different. Are they really significant?”

Response: No, there is no difference of *RUNX1* mRNA distribution between WT and CFBF KO cells. In our previous submission, we used “*” to indicate p value>0.05. In this submission, we use “n.s.” to indicate p value>0.05.

- “- The Figure legends in Supp 2 are not accurate.
- Text mislabels figures 3c and 3d incorrectly—as 5c and 5d.
- Text should mention why the transcripts they validated further were chosen.”*

Response: We thank the reviewer for pointing out these. We have now modified text.

“- What are the relative levels of CFBF and HNRNPK in the HeLa lysate that were used in the in vitro transcription assay? Are they comparable to 293T cells? Is a significant portion of HNRNPK localized in the cytoplasm in this cell line also? Immunofluorescence could be used.”

Response: We examined the levels of CFBF and hnRNPK in HeLa and 293T cells (Supplementary Figure 7e). The levels of CFBF and hnRNPK in HeLa lysate are comparable with those in 293T cells. In both HeLa and 293T cells, a significant portion of hnRNPK was localized in the cytoplasm (Supplementary Figure 7f).

“- On Page 7, the authors state that CFBF does not regulates RUNX1 mRNA degradation. However, while RNA levels are not altered in their experiments, this manuscript does not present data showing RNA degradation or RNA half-life studies. This is particularly relevant, since the authors show that CFBF engages with an RNA binding protein at the 3'UTR of the RUNX1, and the 3'UTR of many mRNA often dictate stability. Thus to be accurate, and without any type of actinomycin studies it may be difficult to definitively state that degradation it not impacted.”

Response: We performed experiment using actinomycin D treatment to measure the half-life of *RUNX1* mRNA. The result showed that CFBF does not regulate the half-life of *RUNX1* mRNA (Supplementary Figure 3h).

“- On page 10, the authors refer to segment T14 of the RUNX1 3'UTR as having enhancer activity. It enhances translation but is not an enhancer per se. Although minor, it would be useful to amend this statement since RUNX1 levels are controlled by enhancer elements and the original statement may cause confusion.”

Response: We “borrowed” the concept of enhancer from the transcription field to emphasize that T14 enhances the translation of *RUNX1*. The reviewer is correct. This may cause confusion of the readers. In this submission, we have modified text to avoid confusion.

[REDACTED]

Figure 2 for reviewers

Figure 2 for reviewers. IF showing the subcellular localization of CBFB in WT and RUNX1 KO MCF10A cells

REVIEWERS' COMMENTS:

Reviewer #1 (Remarks to the Author):

The authors addressed the main concerns of this reviewer.
Please revisit the section between lines 235-276 as some of the sentences are unclear.

Reviewer #2 (Remarks to the Author):

The authors have addressed my previous critique and concerns to satisfaction and the work is suitable for publication in the journal.

Reviewer #3 (Remarks to the Author):

The authors has made attempts to address all concerned raised by the reviewers. For the majority of the comments, the reviewers experimentally addressed each point. For several others, the authors provide explanation for why this was not feasible or the data in question could not be obtained.

In doing so, the authors have addressed all major concerns.

REVIEWERS' COMMENTS:

Reviewer #1 (Remarks to the Author):

The authors addressed the main concerns of this reviewer.

Please revisit the section between lines 235-276 as some of the sentences are unclear.

Response: We have modified some of the sentences between lines 235-276 to increase clarity.

Reviewer #2 (Remarks to the Author):

The authors have addressed my previous critique and concerns to satisfaction and the work is suitable for publication in the journal.

Reviewer #3 (Remarks to the Author):

The authors has made attempts to address all concerned raised by the reviewers. For the majority of the comments, the reviewers experimentally addressed each point. For several others, the authors provide explanation for why this was not feasible or the data in question could not be obtained.

In doing so, the authors have addressed all major concerns.